# The Impact of Blockchain Technology Adoption on an E-Commerce Closed-Loop Supply Chain Considering Consumer Trust

**Deqing Ma, Pengcheng Ma** 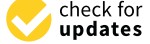 **and Jinsong Hu ***

School of Business, Qingdao University, Qingdao 266071, China; madeqing@qdu.edu.cn (D.M.); 2021021183@qdu.edu.cn (P.M.)
* Correspondence: hujinsong@qdu.edu.cn

**Abstract:** This paper analytically explores the value of blockchain technology in building consumer trust in recyclers. We focus on an e-commerce closed-loop supply chain composed of an online platform and a manufacturer. In the forward chain, the platform selects a reselling or marketplace model to sell products. In the reverse chain, the platform collects used products, and the unknown whereabouts of the used products will cause consumer mistrust and be detrimental to the corporate image. Blockchain technology can address these challenges by improving the visibility of the recycling chain. By constructing differential game models, we specify the conditions for blockchain implementation and explore its impact on the online sales model choice and the E-CLSC performance. The findings show that the manufacturer consistently benefits from blockchain technology, while the platform decides to adopt it when the long-term profits outweigh the initial investment costs. Interestingly, the sales model selection will not change with the advent of blockchain technology. We further show the benefits of blockchain-enabled recycling and provide tangible insights for related practitioners.

**Keywords:** e-commerce closed-loop supply chain; blockchain technology; consumer trust; online platform; model selection; differential game theory

## 1. Introduction

The emergence of online platforms has not only revolutionized the traditional forward supply chain but has also driven a dramatic change in the reverse recycling chain [1,2]. In this context, many online recycling platforms have emerged worldwide, such as Gazelle (San Diego, CA) in the USA, FLIP4NEW in Germany, and Aihuishou in China. With the support of online platforms, information, logistics, and capital flows can be traded electronically in the recycling process, while the concept, technology, and online recycling methods of the Internet are also integrated into the whole process of resource recycling [3]. On these grounds, the e-commerce closed-loop supply chain (hereafter referred to as the E-CLSC), where the online platform is accountable for recycling, is gradually taking shape [4]. E-CLSCs have two critical advantages over traditional closed-loop supply chains, namely the convenience advantage and the technological capability advantage. First, the traditional offline recycling model is constrained by time and space, leading some customers to be reluctant to return used products to a designated location, choosing instead to simply discard them [5]. In contrast, platforms in E-CLSCs use information technology to facilitate communication between recycling practitioners and the general public, providing an accessible channel for residents to participate in recycling [6]. Residents are free to arrange for door-to-door collection or transactions through the platform by appointment without time or space constraints, increasing their convenience and motivation to recycle. For example, a sleepy-eyed person can receive a push of recycling information and submit a waste collection request while checking his or her mobile phone alarm clock. In addition,

technological capability is an important embodiment of E-CLSC strengths. Actually, an online platform aggregates a large amount of valuable user data in its long-term operation and has sophisticated big data analysis. On the one hand, the platform can analyze and understand users' needs according to their behavioral tracks (such as web browsing records, product purchase records, GPS location information, etc.) and conduct big data marketing to improve consumers' favorability and willingness to consume products [7]. On the other hand, the platform can use online advertising and other marketing methods to raise consumers' awareness of environmental protection and educate more users about recycling policies. It can also assess product lifecycles based on transaction information to accurately predict consumers' recycling needs and place recycling messages to guide them to return their used products. As a result, consumers' intentions to recycle have increased, and more and more manufacturers are participating in E-CLSCs [8].

However, in an E-CLSC, the advantages on the recycling side do not mean transparency in the recycling chain, and the ecofriendly recycling practices claimed by the platform and the manufacturer do not always unfold as they should. Specifically, we are concerned about the phenomenon that recyclers face a commitment dilemma in the recycling and remanufacturing of used products. Recyclers often have more convenient and low-cost processing options, such as extracting precious metals through rough refining or other non-environmentally friendly processes [9]. Some recyclers also export hazardous materials, incinerate used products, and even dump waste into the sea, resulting in generating lucrative profits outside the regulatory system and aggravating environmental pollution [10,11]. At this point, for some recyclers, the environmental image they project is often just a "cash cow" for making profits, which leads to the fact that green consumers who are willing to recycle cannot trust them and are unwilling to deliver waste products. According to a recent poll, 63.7% of participants claimed they preferred to keep their used phones at home due to a fear of improper disposal and a lack of trust in recyclers [12]. Similarly, a survey by Echegaray and Hansstein [13] revealed that although most respondents had a positive intention to recycle, only 6% of them actually took action. Inadequate recycling of used products will seriously harm the ecosystem. It is estimated that by 2050, the alarming volume of waste generation will increase to 3.5 billion tons, while only 13% of this volume will be recycled [14]. Behind these appalling figures are the problems of opaque recycling chain processes, lack of consumer trust, and difficulties in building the brand image of recyclers that are currently faced in E-CLSCs. Those sustainable recycling companies that do recycle and remanufacture used products in an environmentally friendly manner find it difficult to gain consumer trust, which hinders the conversion of consumers' recycling intentions into recycling actions. For this reason, there is an urgent need for a means to ensure the transparency and traceability of the recycling process and, thus, the reliability of the sustainability practices of recyclers.

The recent advent of blockchain, a disruptive technology, has offered an opportunity to tackle these problems. Blockchain is a decentralized digital ledger technology with key features such as immutability, visibility, and traceability, designed to improve the efficiency, transparency, and security of various transactions [15,16]. The blockchain-based traceability system can provide all parties involved with an indelible record of the recycling process. Once a consumer has delivered a used product, the system can begin to record and continuously share the entire process of used product disposal, providing a transparent visual representation of the recycling process for all members involved [17]. In practice, some pioneering organizations have already started attempting to use blockchain technology to address recycling issues. For example, Circularise, a blockchain sustainability startup in the Netherlands, has established a blockchain-based platform to foster sustainable development in the plastics industry by ensuring transparent data sharing among recycling participants [18]. Automotive giants such as BMW and Ford have also announced a partnership with the Mobility Open Blockchain Initiative (MOBI) to explore applications such as blockchain recycling in a new digital mobility ecosystem [19]. Meanwhile, considering the transparency and trustworthiness of blockchain technology, some manufacturers, such

as Hewlett-Packard in California, USA, H&M in Vsters, Sweden, and Calik Denim in Istanbul, Turkey, are gradually adopting blockchain to disclose and verify the recycling information of used products [20]. The traceability feature of blockchain technology can provide consumers with detailed and verified information on the recycling of used products by scanning the accompanying QR code, and the immutability feature ensures that the recycling information is authentic and reliable [21]. Therefore, recycling with blockchain technology can provide much-needed transparency and credibility to recyclers, which can help reduce information asymmetry between the E-CLSC and consumers and promote consumers' recycling behavior. In addition, another benefit of blockchain that may be overlooked is that recycling using blockchain technology helps with corporate image. A recycling chain enabled with blockchain technology can display the disposal process of used products, indicating that members of the recycling chain are always carrying out recycling and remanufacturing operations instead of adopting environmentally harmful practices, which dispels consumers' concerns while also building a good corporate image.

When considering implementing blockchain technology to enable recycling for the E-CLSC, it becomes an important issue to explore the shock of blockchain technology on the sales model. As a key component of the E-CLSC, the online platform sales model is an essential strategic choice [22,23]. Generally, there are two models for selling products online: the reselling model and the marketplace model [24]. In the reselling model, the manufacturer wholesales the product to the platform, which then sells it to consumers at a retail price. In the marketplace model, the manufacturer can sell the product directly to the consumer through the platform but must pay a percentage commission to the platform. These two sales models are widely available in the current e-commerce environment. For example, JD derives its primary revenue from the reselling model, while Tmall.com and Taobao.com operate mainly under the marketplace model [25]. Kaplan [26] pointed out that the advantages of emerging technologies can only be unleashed when combined with a business model that allows companies to capture the ultimate value. Therefore, one may question whether blockchain technology will adapt to existing sales channels. Specifically, in an E-CLSC, adopting new technologies may cause changes in various decisions that may affect the choice of platform sales model. Consequently, we highlight the effect of blockchain on the sales model selection.

Although the continuous advancement of blockchain technology and its promised benefits have attracted the attention of many industry personnel and academics, deploying blockchain technology in the E-CLSC requires the coordination of each member's activities (decisions and strategies). In addition, the transparency and trustworthiness that enable recycling with blockchain technology also mean incurring deployment costs. How the trade-off between the cost of blockchain deployment and gaining consumer trust will be made, how the adoption of a blockchain-enabled recycling system will affect members' decisions, what impact blockchain technology will have on the sales model choice, and who will benefit from blockchain implementation are open questions. These answers will shape the degree of acceptance of the technology in the E-CLSC. To this end, the following research questions are explored in the context of the E-CLSC: (1) Under what conditions will the online platform use blockchain technology to enable recycling? (2) Will implementing blockchain technology affect the platform's sales model selection and the cooperation intentions of the manufacturer? How does the deployment affect the decision-making of each E-CLSC member? (3) Under what conditions will blockchain technology improve the E-CLSC performance in terms of triple sustainability (regarding the economy, society, and environment)?

To address these research questions, this paper considers an E-CLSC comprising a manufacturer and an online platform. The manufacturer collaborates with the online platform in sales and recycling. In the sales chain, the online platform offers both reselling and marketplace sales modes, as well as advanced big data marketing services. In the recycling chain, the online platform provides recycling services while deciding on blockchain adoption to enhance the transparency of the recycling chain in order to engender consumer

trust and maintain a positive corporate image. The manufacturer is responsible for the remanufacturing activities of the used products. To capture the dynamic impact of platform marketing activities and consumer trust on brand goodwill and demand, we establish differential game models for different scenarios based on platform's sales model and whether to invest in blockchain technology-enabled recovery. Then, the model is solved using optimal control theory, and the analytical results are analyzed by comparative analysis and numerical examples.

The main contributions of our work are summarized as follows: (1) Existing research regarding the use of blockchain in the E-CLSC has only arisen in the last few years and is still in the early stages of development. A few studies have developed game theoretical models to address the challenges that have emerged in the E-CLSC through blockchain technology. Our work adds to this research by analyzing the effect of blockchain adoption on the E-CLSC through the development of differential game models from a dynamic perspective. (2) We quantitatively examine the value of blockchain-enabled recycling in engendering consumer trust in the E-CLSC. The results suggest that the manufacturer will always benefit from implementing blockchain technology, while the platform may become worse in some cases. In addition, the study found that the improvement of using blockchain-enabled recycling hinges on the level of trust consumers have in the recyclers. Blockchain technology will create more value when trust is low, in which case it is more effective in improving the economic effects, social welfare, and environmental benefits of the E-CLSC. (3) Considering the various advantages of the platform on sales and recycling, we provide important insights about the platform power in the sales process and internalize the big data marketing services into the platform's decisions. In the recycling process, the recycling service level is internalized into the platform's decision. Different from the previous literature [27,28], we modify the reverse recycling function to relate to the platform recycling service level and consumer trust level. The platform recycling service level represents the level of convenience that consumers can enjoy when participating in online recycling, which determines consumers' intentions to recycle [5,29]. Correspondingly, consumer trust level represents the degree of trust that consumers have in the ongoing behavior of the recycling parties, which determines the extent to which their recycling intentions translate into recycling actions. Capturing consumer recycling behavior in this way is more in line with reality. (4) We investigate whether the advent of blockchain technology will change the sales model selection of E-CLSC members. Surprisingly, the results show that blockchain technology does not impact the existing optimal configuration of the sales model and improves the performance of the E-CLSC under certain conditions, which provides decision support for enterprises to adopt blockchain for recycling while maintaining the existing sales model.

The remainder of this paper is structured as follows: Section 2 gives a review of the relevant literature. Section 3 describes our research model and assumptions. Section 4 derives and analyzes the equilibrium outcomes for four scenarios. Section 5 compares the equilibrium outcomes and analyzes the influence of blockchain-enabled recycling on the E-CLSC. Section 6 performs the numerical analysis. Section 7 presents robustness checks. Section 8 concludes our work and provides recommendations for future research. All proofs are in the Appendix A.

## 2. Literature Review

There are three streams of literature closely related to our research: strategic decisions of E-CLSC, blockchain technology in operations and supply chain management, and sales model selection for online platforms.

### 2.1. Strategic Decisions of E-CLSC

The concept of E-CLSC is receiving increasing attention from researchers with the development of online platforms. A platform is a type of two-sided market that combines technical and business capabilities and helps parties interact and collaborate [30]. As an

essential element of the E-CLSC, the online platform not only brings an online recycling channel different from the traditional recycling channel but also plays an ongoing role in achieving value co-creation, enhancing ecological resilience, and promoting sustainable development by taking advantage of technological advantages and network effects [31]. Feng et al. [32] first introduced the concept of online recycling into a traditional recycling chain. They proposed a profit-sharing contract and a two-part tariff contract to coordinate the recovery system. Li et al. [33] similarly noted the rapid development of Internet technology and researched the impact of introducing an online recycling channel in a setting of stochastic demand. They found that online recycling is beneficial to the remanufacturer but detrimental to the recycler. Xiang and Xu [34] examined the effects of big data influence, technological innovation, and overconfidence on the E-CLSC members' decisions based on differential game theory. They showed that an appropriate cost-sharing ratio can be a "win-win" for the manufacturer and the Internet recycling platform, but overconfidence harms the manufacturer's profits. Gong et al. [35] considered the impact of a deposit refund system on a platform-led E-CLSC. The results showed that implementing a deposit refund system always benefits the online platform economically and contributes to environmental benefits. Zhang et al. [36] analyzed who should lead recycling in the setting of regulatory pressure and technological innovation from the perspectives of environmental benefits, economic benefits, and social welfare. They found that online platform-led recycling is optimal in terms of social welfare and environmental benefits, while manufacturer-led recycling performs best economically when fixed costs are appropriate. Wang et al. [37] explored the effects of government regulation and altruistic preferences on an E-CLSC. Their research showed that government regulation and altruistic preferences contribute to the level of recycling services, quality improvement, and recycling quantity. Matsui [38] considered competitive factors in a recycling chain. He discovered that recycling firms can gain a first-mover advantage by announcing prices in advance through the online recycling channel.

As the research progressed, some scholars began to explore the demand side, that is, the factors by which consumers participate in online recycling, which is favorable for the sustainable development of the E-CLSC. According to the theory of planned behavior (TPB), Wang et al. [29] explored residents' willingness to engage in online e-waste recycling and the factors influencing it. The results showed that convenience is the most significant advantage in stimulating residents to participate in recycling through an online platform. Shen et al. [39] investigated the factors that influence consumers to return used mobile phones to an online recycling platform. Their research found that improvements in platform recycling services can directly or indirectly facilitate recycling. Tang and Chen [40] examined why consumers resist digital device recycling platforms. They showed consumers' concerns about data security can hinder recycling behavior due to a lack of trust in recyclers. Hsu [41] considered using gamification mechanisms to improve platform recycling efficiency. He discovered that successful gamified website design can help companies promote user need satisfaction and create intrinsic motivation, thus increasing user engagement in resource recovery.

Our research differs in various aspects from their papers, most notably in that we focus on the online platform in an E-CLSC that accommodates both sales and recycling, where the platform is the decision maker for both product sales and used product recycling. In addition, we consider that the opacity of the recycling process leads to consumer mistrust, which discourages consumer participation in recycling and damages corporate goodwill. On this basis, we attempt to test the role of blockchain technology in solving the problem by developing a stylized theoretical model.

### 2.2. Blockchain Technology in Operations and Supply Chain Management

As one of the most popular disruptive technologies, the recent rise of blockchain technology has garnered the attention of academics in operations and supply chain management. Babich and Hilary [16] comprehensively discussed the advantages and disadvan-

tages of blockchain in operations management. They point out the prospect of blockchain in tracking reverse recycling logistics to verify recycling processing information. Dutta et al. [42] indicated the technological features of blockchain would transform supply chain operations and analyzed its potential for reform in industrial sectors such as manufacturing, energy, technology, and e-commerce. Using the cobalt mining and pharmaceutical industries as examples, Hastig and Sodhi [43] examined the business requirements and success factors for applying blockchain to achieve supply chain traceability. They highlighted that blockchain technology can transparently display supply processes and track product footprints, improving supply chain operational efficiency, sustainability, and the control of illegal practices. Wang et al. [44] explored the benefits and challenges of blockchain technology in supply chain practices. They identified improved supply chain visibility, secure information sharing, and trust building as the expected benefits of using blockchain technology. Considering the complexity and low credibility of traditional diamond identification, Choi [45] investigated the blockchain application in diamond industry. He elucidated the applicable scenarios of the blockchain technology supported platform. Yoon et al. [46] examined the effectiveness of blockchain technology in a global supply chain. They showed that blockchain enables the firm to respond to demand fluctuations more effectively. Iyengar et al. [47] explored the incentives for permissioned blockchain adoption in the supply chain and related industries. They discovered that blockchain adoption improves consumer welfare, but whether it benefits society depends on blockchain adoption costs. In the food industry, Dong et al. [48] examined how the traceability of blockchain technology can be used to control and prevent food contamination. Their research revealed that the performance of blockchain is influenced by the structure of the supply chain network. Tse et al. [49] observed the problems in the Chinese food supply chain and analyzed a supply chain system platform based on blockchain technology. They indicated that promoting blockchain technology can benefit consumers, manufacturers, and regulators. In addition, the combination with the Internet of Things (IoT) is one of the latest application trends in blockchain technology. Rathee et al. [50] proposed a framework for industrial IoT based on blockchain technology, which significantly reduces the rate of product loss, black hole attacks, and falsification among different nodes. Similarly, Cao et al. [51] developed a blockchain-based IoT quality traceability system for steel. The results showed that the system can effectively solve the problems of low transparency and incomplete information in the traditional information process.

However, as a type of supply chain, the blockchain application in the recycling chain (reverse supply chain) has received less attention, which indicates that the field is still in its early stages. Saberi et al. [52] investigated the blockchain application in sustainable supply chain management in terms of economic, environmental, and social dimensions. They pointed out the technical potential of blockchain in enabling recycling, such as product tracking, verifying processing information, and facilitating recycling. In the context of Industry 4.0, Esmaeilian et al. [17] proposed four functions of blockchain that contribute to recycling, including green behavior motivation, product lifecycle visualization, operational efficiency improvement, and sustainability monitoring. Chidepatil et al. [53] developed the first smart contracts enabled by multi-sensor artificial intelligence tools. They demonstrated how these smart tools can help with recycle waste in the plastics industry. Gopalakrishnan et al. [54] constructed an optimization model that refined the cost of solid waste disposal and expressed the blockchain cost as an initial fixed cost for evaluating the cost of a recycling system based on a blockchain technology platform. Howson [55] discussed the potential of blockchain technology to protect the marine environment, such as by promoting transparency in marine conservation, reducing plastic pollution, and ensuring sustainable management of fisheries.

According to the above research, the discussion of blockchain-enabled recycling has only emerged in the last few years. Few scholars have focused on developing game theoretical models to leverage the advantages of blockchain to address the challenges that have arisen in the E-CLSC. Our research adds to this intriguing topic by examining the value

of blockchain technology in an E-CLSC through differential game theory, where blockchain technology is used to enhance the visibility of the used product recycling process, gain consumer trust, and maintain the corporate image.

### 2.3. Sales Model Selection for Online Platforms

Choosing the appropriate online sales model is one of the issues that cannot be ignored by online platforms and has been widely discussed by scholars recently. Two historically dominant sales models are the marketplace (agency) and reselling models. Hagiu and Wright [24] examined the optimal positioning selection between the above two sales models by establishing some basic trade-offs. They found that controlling for noncontracting decision variables such as marketing activities plays a crucial role in the selection. Abhishek et al. [56] addressed the optimal sales model for two competing online platforms given the existence of a traditional offline channel. The results showed that the sales model selection depends on the spillover effect between the electronic and traditional channels and the intensity of competition among e-retailers. Considering the competition among suppliers, Tian et al. [57] discussed an online platform offering sales to two competing suppliers. Their research found that competition intensity and fulfillment costs determine the sales model selection. Kwark et al. [58] also investigated the issue of sales model selection for online platforms, but they focused on the impact of consumer reviews. Chen et al. [59] examined the effect of customer loyalty on selecting an online sales model. They revealed that the marketplace model outperforms the reselling model only when customer loyalty is strong enough. Yan et al. [60] researched the incentives for a manufacturer and an e-tailer to implement a marketplace channel. They showed that selling through the marketplace channel is feasible even when the manufacturer has disadvantages in terms of sales efficiency and demand information. In the context of generalized Nash bargaining, Shen et al. [23] considered the situation in which a manufacturer and an online platform bargain over commission rates and entrance fees. The results showed that the manufacturer should engage with the platform under specific demand substitution values. Qin et al. [61] combined the logistics service strategy and the sales model selection to investigate the key role played by the cost performance of logistics services. Pu et al. [62] explored how the manufacturer chooses a sales model when channel operating costs are present. They indicated that operating costs and commissions determine the optimal online sales model. Jia and Li [8] investigated sales model choice from four perspectives: vendor, platform, consumer, and environmental. They revealed that platform fees and order fulfillment costs are essential factors influencing distribution channel preferences, but they ignored the impact of platform marketing services. Subsequently, Ha et al. [25] examined the impact of platform promotion services on sales channel selection. The results suggested that the channel flexibility brought by dual channels may create more service effort motivation for the platform. Xu et al. [63] researched the sales model selection in the context of a cap-and-trade system. They found that demand disruption affects the manufacturer's preferences for sales models. Similarly, in the background of cap-and-trade regulation, Xu and Choi [64] examined the strategic selection of reselling and marketplace models. They showed that the selection of a sales model is correlated with commission rates and elucidated the specific situations in which blockchain can create value. Wei and Dong [65] explored the incentive for an online platform to introduce a marketplace channel and the optimal sales model choice for a supplier. They showed that the level of product differentiation and order fulfillment costs play a vital role in determining the sales model selection. Zhang and Hou [66] observed the phenomenon of private labels introduced by e-commerce platforms. Their research indicated that when the platform owns private label products, the sales model preferences of the manufacturer and the e-commerce platform are opposite in most cases. They further demonstrated that the finding still holds when asymmetric information about production and sales costs exists.

Different from the existing literature, our study has several distinctive features. First, we focus on different online sales models (marketplace and reselling models) and investi-

gate the decisions of an online platform and a manufacturer in the presence of a reverse supply chain. In contrast, the aforementioned literature focuses only on the choice of a forward sales model and does not consider the role of platforms in facilitating recycling, ultimately simplifying sales model dynamics. Second, our research is among the few to model the problem of using blockchain technology to win consumer trust in recyclers, given that consumer trust in the entire E-CLSC affects recycling and corporate goodwill. Finally, we shed light on the influence of blockchain-enabled recycling on the sales model selection of the platform and the cooperation intentions of the manufacturer from a dynamic perspective. In this light, we explore the influence of transparency and trustworthiness brought by blockchain technology on the triple performance of the economy, society, and environment. Table 1 compares the studies that are most relevant to this paper.

**Table 1.** Comparison of the most relevant studies with this paper.

| References | E-CLSC | Consumer Trust | Blockchain Technology | Online Sales Model Choice |
|:---:|:---:|:---:|:---:|:---:|
| [34] | √ | | | |
| [35] | √ | | | |
| [37] | √ | | | |
| [40] | | √ | | |
| [16] | | | √ | |
| [42] | | | √ | |
| [43] | | | √ | |
| [48] | | | √ | |
| [44] | | √ | √ | |
| [45] | | √ | √ | |
| [23] | | | | √ |
| [21] | | | | √ |
| [56] | | | | √ |
| [57] | | | | √ |
| [58] | | | | √ |
| [64] | | | √ | √ |
| Our paper | √ | √ | √ | √ |

## 3. Model Description and Assumptions

We consider an E-CLSC composed of a manufacturer (M) and an online platform (P). Note that the opaqueness of the recycling process can lead to consumer mistrust, which hinders the shift from consumer recycling intentions to recycling actions. The adoption of blockchain technology can provide a transparent visual recycling process for all parties involved [17], thus ensuring the credibility of the ongoing behavior of recycling parties and promoting consumer recycling behavior. In the recycling process, we considered scenarios with and without blockchain-enabled recycling. Two common sales models considered in the sales process are the reselling and marketplace models [24,56].

In the recycling chain, the online platform can offer recycling services to consumers at their preferred time, significantly increasing the convenience of consumer participation in recycling. Moreover, the platform also has a large amount of high-quality user data from its long-term operations and has advanced big data service technology, which can accurately predict consumer recycling demand and place targeted advertising to encourage consumers to return used products. The recycling services provided by the platform, such as door-to-door recycling and big data services, are expressed in terms of the recycling effort $u(t)$. Based on the above description, the recycling quantity of used products can be expressed as follows:

$$D_b = (a + \varepsilon u(t))\xi \tag{1}$$

where $a > 0$ is the basic scale of recycling, $\varepsilon > 0$ is the effectiveness of the recycling service effort to enhance the recycling intentions of consumers, and $0 < \xi < 1$ is the "trust factor" which is less than 1, reflecting the level of consumer trust in the sustainable behavior of the

recyclers. A larger $\xi$ represents a higher level of trust. Some investigations have pointed out that the convenience of online recycling is the main factor affecting consumers' intentions to participate in online recycling [29,67]. Therefore, in the portrayal of the reverse recycling function, we split the process of consumer participation in recycling into two stages, namely the formation of the intentions to recycle in the first stage and the realization of the recycling actions in the second stage. $(a + \varepsilon u(t))$ represents the consumers' intentions to recycle in the first stage, which is facilitated by the convenient recycling service experience provided by the platform. In the second stage, the realization of consumer recycling actions depends on the level of consumer trust in the recyclers. In other words, in the absence of credible proof of the recyclers' sustainable behavior, $(1 - \xi)$ proportion of consumers will refuse to participate in recycling, so the final quantity of recycling is achieved in $D_b$. The cost of the platform investment in the recycling effort is assumed to be a convex function of a quadratic form and given by

$$C(u(t)) = \frac{1}{2}k_u u(t)^2 \tag{2}$$

where $k_u > 0$ is the cost parameter for the platform to conduct the recycling effort. The platform passes the collected used products to the manufacturer, who receives a marginal economic return $\Delta$ from the used product through remanufacturing. To incentivize the platform to engage in recycling activities, the manufacturer shares in the recycling revenue [68]. We assume that the platform shares the recycling benefit through a reverse revenue sharing contract (RRSC). $0 < \alpha < 1$ represents the recycling benefit sharing rate between the platform and the manufacturer and is exogenously given, which means that the platform shares the marginal benefit of $\alpha\Delta$ from recycling and remanufacturing a used product while the manufacturer retains the unit benefit of $(1 - \alpha)\Delta$.

Moreover, we assume that consumer mistrust will be detrimental to the accumulation of brand goodwill. Concretely, from the consumers' perspective, an opaque recycling process implies the possibility of improper waste disposal, which is inconsistent with the original vision of consumer participation in recycling, i.e., the desire to put used products into remanufacturing production to save resources and protect the environment. To improve brand goodwill, the online platform undertakes marketing efforts $s(t)$ during the sales process, including the segmentation of customers based on user behavior and the provision of big data targeted advertising [69]. To capture the impact of consumer trust and marketing effort on brand goodwill, we extend Nerlove–Arrow's equation [70], as described in the following dynamics equation:

$$\dot{G}(t) = \rho s(t) - (\delta + 1 - \xi)G(t), \quad G(0) = G_0 \geq 0 \tag{3}$$

where $\delta > 0$ is the forgetting effect of brand goodwill; $\rho \geq 0$ is the effectiveness of the marketing effort; and $(1 - \xi) \geq 0$ is the level of consumer mistrust of the recyclers, which exacerbates the decay of brand goodwill. $G_0 \geq 0$ is the initial goodwill. Similar to the cost of recycling effort, the marketing cost is expressed in a standard convex function, including the cost of investing in technology development, advertising placement, and big data marketing, as expressed by the following function:

$$C(s(t)) = \frac{1}{2}k_s s(t)^2 \tag{4}$$

where $k_s > 0$ is the cost parameter for the platform to carry out its marketing effort. The convex function explains that the marketing cost function conforms to the law of diminishing marginal returns.

In the sales process, the platform can cooperate with the manufacturer under different sales models. Specifically, in the reselling model, the manufacturer decides the wholesale price $w(t)$, and the online platform sets the retail price $p(t)$. In the marketplace model, the platform allows the manufacturer to sell the product directly to the consumer but will charge a commission at a rate of $\varphi$; the manufacturer determines the retail price $p(t)$. For

the size of the commission rate $\varphi$, we assume that $\varphi$ is exogenously given and determined by the product category. In practice, the commission rate is predetermined by the category of products. For example, Amazon.com charges 17% for clothing and accessories, 5% for office supplies, and 8% for computers [71]. Moreover, we assume that the demand level is influenced by sales price and brand goodwill, and the consumer demand function is described by the following equation:

$$D_f = (\theta + \lambda)\sqrt{G(t)} - \beta p(t) \tag{5}$$

where $\theta > 0$ represents the sensitivity of consumers to brand goodwill. $\theta\sqrt{G(t)}$ represents the basic market size. Here, we assume a square root function, which suggests that demand does not expand indefinitely as brand goodwill increases, i.e., there is a saturation effect [72]. As a multilateral marketplace that incorporates technical competences and business capacities, the platform provides an environment for the participants to interact and create value, making it easier for the companies in it to gain a competitive advantage [30]. Therefore, we use $\lambda > 0$ to represent the key resources (e.g., number of users) and core capabilities (e.g., technical and business capabilities) owned by the platform, referred to as platform power. $\lambda\sqrt{G(t)}$ represents the expansion of platform power to the manufacturer's potential market. $\beta > 0$ is the consumer sensitivity to retail price and denotes the number of consumers lost for each unit increase in the retail price.

When the platform implements blockchain technology for recycling, the traceability feature of blockchain technology provides consumers with a transparent and visual recycling process [17]. Although the platform and the manufacturer may have priority access rights, such as data entry, the data-invariant nature of the blockchain ensures that the recycling information is trustworthy. Specifically, in such a 100% decentralized system, members of the recycling chain would be required to write immutable data about their recycling and remanufacturing activities on the blockchain. This decentralized system eliminates the possibility of a single node in a conventional centralized system deliberately concealing or tampering with data. At this point, blockchain technology provides the power to rebuild the conventional centralized recycling chain system because employing the blockchain-based decentralized mechanism can provide visible recycling information and create a trusted recycling environment. We let $\xi = 1$ to represent the highest level of consumer trust in recyclers when deploying the blockchain. Then, the recycling function and the brand goodwill dynamics equation are given as follows:

$$D_b = a + \varepsilon u(t) \tag{6}$$

$$\dot{G}(t) = \rho s(t) - \delta G(t), \quad G(0) = G_0 \geq 0 \tag{7}$$

We assume that the blockchain in this paper is a permissioned blockchain and that the cost of using the blockchain is the initial investment cost $F$ [73]. In Section 7, we extend the model and consider the case of partnering with a blockchain technology provider, when the cost of using the blockchain is the unit cost $b$ incurred per use [74].

Without loss of generality, we normalize the production cost to zero. We assume that the E-CLSC members aim to maximize profits and play the game with a discount factor $r$ over an infinite period, where $0 \leq r \leq 1$. This game is a Stackelberg game where the manufacturer is the leader. In practice, it is not uncommon for manufacturers to act as channel leaders in E-CLSCs. For example, in 2020, Apple planned to partner with JD.com and Aihuishou to promote the reuse of used products. Clearly, Apple has enough channel power to lead such E-CLSCs when collaborating with online platforms. Table 2 displays the meanings of all the notations in the research.

**Table 2.** Nomenclature/Abbreviations.

| | Notations |
|---|---|
| $a$ | The basic scale of recycling |
| $\varepsilon$ | Recycling effectiveness of the online platform |
| $\zeta$ | Consumer trust in recyclers |
| $\Delta$ | Marginal revenue of used products |
| $\alpha$ | Recycling revenue sharing rate |
| $\delta$ | Forgetting effect |
| $\rho$ | Marketing effectiveness of the online platform |
| $\varphi$ | Commission rate for products |
| $\theta$ | Consumer sensitivity to brand goodwill |
| $\lambda$ | Platform power |
| $\beta$ | Consumer sensitivity to retail price |
| $r$ | Discount factor |
| $F$ | Blockchain initial fixed investment cost |
| $b$ | Unit cost of using blockchain technology |
| $k_s$ | Cost parameter of online platform marketing effort |
| $k_u$ | Cost parameter of online platform recycling effort |
| $G_0$ | Initial brand goodwill |
| $\tau$ | Recycling rate |
| $E_e$ | Economic efficiency of the E-CLSC with blockchain technology |
| $E_s$ | Social efficiency of the E-CLSC with blockchain technology |
| $E_g$ | Environmental efficiency of the E-CLSC with blockchain technology |
| $D_f$ | Consumer demand |
| $D_b$ | Quantity of used products recycling |
| *State Variable* | |
| $\dot{G}(t)$ | Stock of brand goodwill |
| *Decision Variables* | |
| $w(t)$ | Wholesale price of products |
| $p(t)$ | Retail price of products |
| $s(t)$ | Online platform marketing effort |
| $u(t)$ | Online platform recycling effort |

Finally, to make our analysis nontrivial, we give the following assumptions:

(1) $D_f > D_b$, which indicates that consumers are unlikely to return 100% of their products to an online platform.

(2) The condition $\Delta < w$ needs to be satisfied in the reselling model and $\Delta < (1 - \varphi)p$ in the marketplace model, which points out that the residual value of the used product cannot be greater than the initial value.

## 4. Model Analysis

In this section, differential game models are developed in different scenarios to derive the equilibrium strategies of E-CLSC members. As mentioned in Section 3, the four scenarios explored in this section are: the reselling model without blockchain-enabled recycling, the marketplace model without blockchain-enabled recycling, the reselling model with blockchain-enabled recycling, and the marketplace model with blockchain-enabled recycling. We use the superscripts *NR*, *NM*, *BR*, and *BM* to denote the four scenarios above, respectively. The online platform and the manufacturer are respectively represented by the subscripts *P* and *M*. For ease of presentation, the calculation procedure will omit the time *t*. The detailed proofs are summarized in the Appendix A. Figure 1 illustrates the four scenarios explored in this paper.

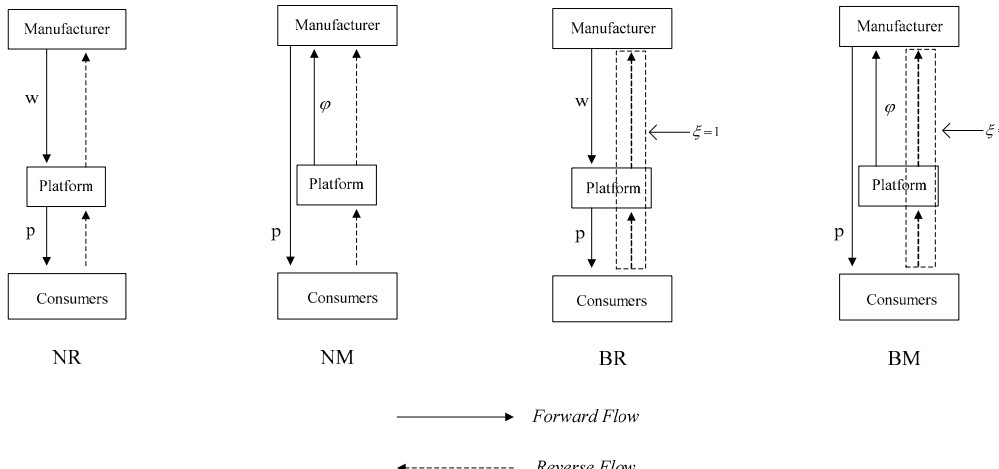

Forward Flow

- - - - - - - > Reverse Flow

**Figure 1.** Four scenarios of the E-CLSC.

*4.1. Scenario NR*

In scenario $NR$, the online platform recycles used products without blockchain and sells products through a reselling model. The game develops as the following steps: the manufacturer first determines the wholesale price $w$; the platform then sets the retail price $p$, the marketing effort $s$, and the recycling effort $u$. Thus, the optimization problem of scenario $NR$ can be summarized as follows:

$$
\max_{w}\left\{ J_M^{NR} = \int_0^\infty e^{-rt}\left[ w(t)D_f^{NR} + (1-\alpha)\Delta D_b^{NR} \right]dt \right\}
$$
$$
s.t. \begin{cases} \max_{p,u,s}\left\{ J_P^{NR} = \int_0^\infty e^{-rt}\left[ (p(t)-w(t))D_f^{NR} + \alpha\Delta D_b^{NR} - \tfrac{1}{2}k_u u(t)^2 - \tfrac{1}{2}k_s s(t)^2 \right]dt \right\} \\ \dot{G}(t) = \rho s(t) - (\delta+1-\xi)G \quad G(0) = G_0 \end{cases} \tag{8}
$$

**Theorem 1.** *In scenario $NR$, the optimal trajectory of the brand goodwill over time is $G^{NR} = \left(G_0 - G_\infty^{NR}\right)e^{-(\delta+1-\xi)t} + G_\infty^{NR}$, where $G_\infty^{NR} = \dfrac{\rho^2(\theta+\lambda)^2}{16\beta k_s(r+\delta+1-\xi)(\delta+1-\xi)}$, $G_\infty^{NR}$ is the steady-state brand goodwill level.*

The E-CLSC members' equilibrium strategies are $w_\infty^{NR} = \dfrac{(\theta+\lambda)\sqrt{G_\infty^{NR}}}{2\beta}$, $p_\infty^{NR} = \dfrac{3(\theta+\lambda)\sqrt{G_\infty^{NR}}}{4\beta}$, $s^{NR} = \dfrac{\rho(\theta+\lambda)^2}{16\beta k_s(r+\delta+1-\xi)}$, and $u^{NR} = \dfrac{\alpha\Delta\xi\varepsilon}{k_u}$; the profits of the manufacturer and the online platform are given by $V_M^{NR} = l_1^{NR}G_\infty^{NR} + l_2^{NR}$, $V_P^{NR} = l_3^{NR}G_\infty^{NR} + l_4^{NR}$, respectively.

Consumer surplus is calculated as $CS_\infty^{NR} = \dfrac{\rho^2(\theta+\lambda)^4}{512rk_s\beta^2(r+\delta+1-\xi)(\delta+1-\xi)}$, social welfare is written as $SW_\infty^{NR} = CS_\infty^{NR} + V_M^{NR} + V_P^{NR}$, where the specific expression for $l_i^{NR}$, $i \in \{1, 2\}$, and the detailed analysis is given in the Appendix A.

According to the above equilibrium outcomes, the impact of some exogenous parameters on E-CLSC members' equilibrium strategies and brand goodwill in scenario $NR$ is presented in Table 3.

Table 3 illustrates the impact of some exogenous parameters on E-CLSC members' equilibrium strategies and brand goodwill in scenario $NR$. As a direct factor of demand (or product sales), an increase in platform power $\lambda$ and consumer sensitivity to brand goodwill $\theta$ means that the products' market response improves. At this time, enterprises are committed to raising the price to obtain more profits. Also, popular products will trigger the platform to engage in more aggressive marketing activities, thus increasing brand goodwill. Conversely, consumer sensitivity to retail price $\beta$ negatively affects the pricing level of products, the brand goodwill, and the effort level of marketing. The better the platform marketing effectiveness $\rho$ is, the faster the brand goodwill will improve. At this point,

consumers are more receptive to the product, which means that both the manufacturer and the online platform gain higher bargaining power and can thus increase the wholesale and retail prices. This suggests that marketing tools can mitigate the adverse impact of raising prices on demand. Comparatively, the marketing cost parameter $k_s$ and the consumer forgetting effect $\delta$ adversely affect the E-CLSC members' equilibrium strategies and brand goodwill. In addition, when the discount rate $r$ increases, the accumulation of brand goodwill becomes a challenging goal to achieve. The significant discount on future earnings hinders the investment in marketing services and lowers the pricing level of products. In the recovery process, the online platform recycling effort is only related to three parameters: the consumer trust $\xi$, the recycling effort cost parameter $k_u$ and the recycling revenue sharing rate $\alpha$. Specifically, the consumer trust $\xi$ and the recycling benefit sharing rate $\alpha$ have a positive impact on the platform recycling effort, while the recycling effort cost parameter $k_u$ is detrimental to the platform recycling effort investment. Finally, we find that an increase in consumer trust $\xi$ will comprehensively enhance the brand goodwill, brand premium capacity, and the effort level of E-CLSC members.

**Table 3.** Sensitivity analysis in scenario $NR$.

| | $\rho$ | $\theta$ | $\lambda$ | $\delta$ | $\beta$ | $r$ | $k_s$ | $\xi$ | $k_u$ | $\alpha$ |
|---|---|---|---|---|---|---|---|---|---|---|
| $G_\infty^{NR}$ | ↗ | ↗ | ↗ | ↘ | ↘ | ↘ | ↘ | ↗ | — | — |
| $w_\infty^{NR}$ | ↗ | ↗ | ↗ | ↘ | ↘ | ↘ | ↘ | ↗ | — | — |
| $p_\infty^{NR}$ | ↗ | ↗ | ↗ | ↘ | ↘ | ↘ | ↘ | ↗ | — | — |
| $s_\infty^{NR}$ | ↗ | ↗ | ↗ | ↘ | ↘ | ↘ | ↘ | ↗ | — | — |
| $u_\infty^{NR}$ | — | — | — | — | — | — | — | ↗ | ↘ | ↗ |

Note: ↗ denotes positive correlation, ↘ denotes negative correlation, — denotes no correlation.

### 4.2. Scenario NM

In scenario $NM$, the online platform recycles used products without blockchain and sells products through a marketplace model. The game develops as the following steps: the manufacturer first determines the retail price $p$; the platform then decides the marketing effort $s$ and the recycling effort $u$. Thus, the optimization problem of scenario $NM$ can be summarized as

$$\max_p \left\{ J_M^{NM} = \int_0^\infty e^{-rt} \left[ (1-\varphi)p(t)D_f^{NM} + (1-\alpha)\Delta D_b^{NM} \right] dt \right\}$$
$$s.t. \begin{cases} \max_{u,s} \left\{ J_P^{NM} = \int_0^\infty e^{-rt} \left[ \varphi p(t)D_f^{NM} + \alpha\Delta D_b^{NM} - \frac{1}{2}k_u u(t)^2 - \frac{1}{2}k_s s(t)^2 \right] dt \right\} \\ \dot{G}(t) = \rho s(t) - (\delta + 1 - \xi)G(t) \quad G(0) = G_0 \end{cases} \quad (9)$$

**Theorem 2.** *In scenario NM, the optimal trajectory of the brand goodwill over time is* $G^{NM} = (G_0 - G_\infty^{NM})e^{-(\delta+1-\xi)t} + G_\infty^{NM}$, *where* $G_\infty^{NM} = \frac{\varphi\rho^2(\theta+\lambda)^2}{4\beta k_s(r+\delta+1-\xi)(\delta+1-\xi)}$, $G_\infty^{NM}$ *is the steady-state brand goodwill level.*

The E-CLSC members' equilibrium strategies are $p_\infty^{NM} = \frac{(\theta+\lambda)\sqrt{G_\infty^{NM}}}{2\beta}$, $s^{NM} = \frac{\varphi\rho(\theta+\lambda)^2}{4\beta k_s(r+\delta+1-\xi)}$, and $u^{NM} = \frac{\alpha\Delta\varepsilon\xi}{k_u}$; the profits of the manufacturer and the online platform are given by $V_M^{NM} = l_1^{NM}G_\infty^{NM} + l_2^{NM}$, $V_P^{NM} = l_3^{NM}G_\infty^{NM} + l_4^{NM}$, respectively.

Consumer surplus is calculated as $CS_\infty^{NM} = \frac{\varphi\rho^2(\theta+\lambda)^4}{32r\beta^2 k_s(r+\delta+1-\xi)(\delta+1-\xi)}$; social welfare is written as $SW_\infty^{NM} = CS_\infty^{NM} + V_M^{NM} + V_P^{NM}$, where the specific expression for $l_i^{NM}$, $i \in \{1, 2\}$; and the detailed analysis is given in the Appendix A.

According to the above equilibrium outcomes, the impact of some exogenous parameters on E-CLSC members' equilibrium strategies and brand goodwill in scenario $NM$ is presented in Table 4.

**Table 4.** Sensitivity analysis in scenario $NM$.

| | $\rho$ | $\theta$ | $\lambda$ | $\delta$ | $\beta$ | $r$ | $k_s$ | $\xi$ | $\varphi$ | $k_u$ | $\alpha$ |
|---|---|---|---|---|---|---|---|---|---|---|---|
| $G_\infty^{NM}$ | ↗ | ↗ | ↗ | ↘ | ↘ | ↘ | ↘ | ↗ | ↗ | — | — |
| $p_\infty^{NM}$ | ↗ | ↗ | ↗ | ↘ | ↘ | ↘ | ↘ | ↗ | ↗ | — | — |
| $s_\infty^{NM}$ | ↗ | ↗ | ↗ | ↘ | ↘ | ↘ | ↘ | ↗ | ↗ | — | — |
| $u_\infty^{NM}$ | — | — | — | — | — | — | — | ↗ | — | ↘ | ↗ |

Note: ↗ denotes positive correlation, ↘ denotes negative correlation, — denotes no correlation.

Table 4 shows that in scenario $NM$, except for the commission rate $\varphi$, the influence of key exogenous parameters on the brand goodwill and the equilibrium strategies of each enterprise is consistent with that in the reselling model, which supports our follow-up research on the influence of the platform commission rate on the sales model selection. As for the commission rate, we find that when the commission rate increases, the platform will have sufficient funds to carry out the marketing effort, which promotes the accumulation of brand goodwill. Correspondingly, the manufacturer will raise the retail price to alleviate the financial pressure from the high commission rate. In addition, the increase in commission rate does not affect the recovery effort, which explains the analysis results that the recovery effort is the same for the reseller model and the marketplace model when the external conditions do not change. Since the change of other parameters has the same impact on the enterprises as scenario $NR$, the analysis will not be carried out here.

**Corollary 1.** *The sales model selection for the platform and the cooperation intentions of the manufacturer when there is no blockchain-enabled recycling: the subsequent findings are obtained after comparing the optimal profits of E-CLSC members under scenario NR and scenario NM. The platform will go for the reselling model if $0 < \varphi \leq 1/4$ and the marketplace model if $1/4 < \varphi < 1$. The manufacturer will collaborate with the platform in the reselling model if $0 < \varphi < \left(2 - \sqrt{2}\right)/4$ or $\left(2 + \sqrt{2}\right)/4 < \varphi < 1$, while the manufacturer will partner with the platform in the marketplace model if $\left(2 - \sqrt{2}\right)/4 < \varphi < \left(2 + \sqrt{2}\right)/4$. The detailed proofs of Corollary 1 are summarized in Section 5.3.*

*4.3. Scenario BR*

In scenario $BR$, the online platform is equipped with blockchain technology to empower recycling and can provide consumers with authentic and reliable information on the disposal of used products to engender consumer trust. In this case, consumer trust $\xi = 1$, and the platform sells products in a reselling mode. The game develops as the following steps: the manufacturer first determines the wholesale price $w$; the platform then sets the retail price $p$, the marketing effort $s$, and the recycling effort $u$. Thus, the optimization problem of scenario $BR$ can be summarized as follows:

$$\max_{w}\left\{J_M^{BR} = \int_0^\infty e^{-rt}\left[w(t)D_f^{BR} + (1-\alpha)\Delta D_b^{BR}\right]\mathrm{d}t\right\}$$
$$s.t.\begin{cases} \max_{p,u,s}\left\{J_P^{BR} = \int_0^\infty e^{-rt}\left[(p(t)-w(t))D_f^{BR} + \alpha\Delta D_b^{BR} - \frac{1}{2}k_u u(t)^2 - \frac{1}{2}k_s s(t)^2 - F\right]\mathrm{d}t\right\} \\ \dot{G}(t) = \rho s(t) - \delta G(t) \quad G(0) = G_0 \end{cases} \tag{10}$$

**Theorem 3.** *In scenario BR, the optimal trajectory of the brand goodwill over time is $G^{BR} = \left(G_0 - G_\infty^{BR}\right)e^{-\delta t} + G_\infty^{BR}$, where $G_\infty^{BR} = \frac{\rho^2(\theta+\lambda)^2}{16\beta k_s(r+\delta)\delta}$, $G_\infty^{BR}$ is the steady-state brand goodwill level.*

The E-CLSC members' equilibrium strategies are $w_\infty^{BR} = \frac{(\theta+\lambda)\sqrt{G_\infty^{BR}}}{2\beta}$, $p_\infty^{BR} = \frac{3(\theta+\lambda)\sqrt{G_\infty^{BR}}}{4\beta}$, $s^{BR} = \frac{\rho(\theta+\lambda)^2}{16\beta k_s(r+\delta)}$, and $u^{BR} = \frac{\alpha\Delta\varepsilon}{k_u}$; the profits of the manufacturer and the online platform are given by $V_M^{BR} = l_1^{BR}G_\infty^{BR} + l_2^{BR}$, $V_P^{BR} = l_3^{BR}G_\infty^{BR} + l_4^{BR}$, respectively.

Consumer surplus is calculated as $CS_\infty^{BR} = \frac{\rho^2(\theta+\lambda)^4}{512rk_s\beta^2(r+\delta)\delta}$; social welfare is written as $SW_\infty^{BR} = CS_\infty^{BR} + V_M^{BR} + V_P^{BR}$, where the specific expression for $l_i^{BR}$, $i \in \{1, 2\}$; and the detailed analysis is given in the Appendix A.

The impact of exogenous parameters on the equilibrium strategies of E-CLSC members and brand goodwill is consistent with Theorem 1 and will not be discussed here.

*4.4. Scenario BM*

In scenario *BM*, the platform uses blockchain technology to enable recycling and sells products in a marketplace model. The game develops as the following steps: the manufacturer first determines the retail price *p*; the platform then decides the marketing effort *s*, and the recycling effort *u*. Thus, the optimization problem of scenario *BM* can be summarized as follows:

$$\max_p\left\{J_M^{BM} = \int_0^\infty e^{-rt}\left[(1-\varphi)p(t)D_f^{BM} + (1-\alpha)\Delta D_b^{BM}\right]dt\right\}$$
$$s.t.\begin{cases}\max_{u,s}\left\{J_P^{BM} = \int_0^\infty e^{-rt}\left[\varphi p(t)D_f^{BM} + \alpha\Delta D_b^{BM} - \frac{1}{2}k_u u(t)^2 - \frac{1}{2}k_s s(t)^2 - F\right]dt\right\} \\ \dot{G}(t) = \rho s(t) - \delta G(t) \quad G(0) = G_0\end{cases} \quad (11)$$

**Theorem 4.** *In scenario BM, the optimal trajectory of the brand goodwill over time is* $G^{BM} = (G_0 - G_\infty^{BM})e^{-\delta t} + G_\infty^{BM}$, *where* $G_\infty^{BM} = \frac{\varphi\rho^2(\theta+\lambda)^2}{4\beta k_s(r+\delta)\delta}$, $G_\infty^{BM}$ *is the steady-state brand goodwill level.*

The E-CLSC members' equilibrium strategies are $p_\infty^{BM} = \frac{(\theta+\lambda)\sqrt{G_\infty^{BM}}}{2\beta}$, $s^{BM} = \frac{\varphi\rho(\theta+\lambda)^2}{4\beta k_s(r+\delta)}$, and $u^{BM} = \frac{\alpha\Delta\varepsilon}{k_u}$; the profits of the manufacturer and the online platform are given by $V_M^{BM} = l_1^{BM}G_\infty^{BM} + l_2^{BM}$, $V_P^{BM} = l_3^{BM}G_\infty^{BM} + l_4^{BM}$, respectively.

Consumer surplus is calculated as $CS_\infty^{BM} = \frac{\varphi\rho^2(\theta+\lambda)^4}{32r\beta^2k_s(r+\delta)\delta}$; social welfare is written as $SW_\infty^{BM} = CS_\infty^{BM} + V_M^{BM} + V_P^{BM}$, where the specific expression for $l_i^{BM}$, $i \in \{1, 2\}$; and the detailed analysis is given in the Appendix A.

**Corollary 2.** *The sales model selection for the platform and the manufacturer's cooperation intentions when there is blockchain-enabled recycling: the subsequent findings are obtained after comparing the optimal profits of E-CLSC members under scenario BR and scenario BM. The platform will go for the reselling model if $0 < \varphi \le 1/4$ and the marketplace model if $1/4 < \varphi < 1$. The manufacturer will collaborate with the platform in the reselling model if $0 < \varphi < (2 - \sqrt{2})/4$ or $(2 + \sqrt{2})/4 < \varphi < 1$, while the manufacturer will partner with the platform in the marketplace model if $(2 - \sqrt{2})/4 < \varphi < (2 + \sqrt{2})/4$. The detailed proofs of Corollary 2 are summarized in Section 5.3.*

## 5. Comparative Analysis: The Impact of Blockchain-Enabled Recycling on the E-CLSC

In this section, we explore the impact of blockchain-enabled recycling on recycling quantity, brand goodwill, demand, equilibrium strategies, and manufacturer profits by comparing the equilibria in different scenarios. In addition, we identify the conditions for blockchain implementation and investigate the impact of blockchain on the optimal sales model selection.

### 5.1. The Impact of Blockchain-Enabled Recycling on Recycling Quantity, Steady-State Brand Goodwill, Demand, Equilibrium Strategies, and Manufacturer Profits

**Theorem 5.** *(1) The impact of blockchain-enabled recycling on recycling quantity, brand goodwill, demand, equilibrium strategies, and manufacturer profits in a reselling model:* $D_b^{NR} < D_b^{BR}$, $G_\infty^{NR} < G_\infty^{BR}$, $D_f^{NR} < D_f^{BR}$, $w_\infty^{NR} < w_\infty^{BR}$, $p_\infty^{NR} < p_\infty^{BR}$, $s^{NR} < s^{BR}$, $u^{NR} < u^{BR}$, $V_M^{NR} < V_M^{BR}$.

*(2) The impact of blockchain-enabled recycling on recycling quantity, brand goodwill, demand, equilibrium strategies, and manufacturer profits in a marketplace model:* $D_b^{NM} < D_b^{BM}$, $G_\infty^{NM} < G_\infty^{BM}$, $D_f^{NM} < D_f^{BM}$, $w_\infty^{NM} < w_\infty^{BM}$, $p_\infty^{NM} < p_\infty^{BM}$, $s^{NM} < s^{BM}$, $u^{NM} < u^{BM}$, $V_M^{NM} < V_M^{BM}$.

Theorem 5 states that using blockchain technology to recycle solves the recycling difficulty problem in the first place. This is because the transparent traceability of the recycling process eliminates the possibility of improper waste disposal (e.g., crude refining, incineration, etc.) and sends credible signals to consumers, which helps promote consumer recycling behavior and increases the quantity of used products recycled. Furthermore, the adoption of blockchain technology is conducive to the establishment of the corporate image. The openness and transparency of sustainable behavior enhance consumer identification with the company, and the steady-state brand goodwill is subsequently increased. With the improvement of brand goodwill, demand and product bargaining power are increased, which indicates that the deployment of blockchain mobilizes consumers' enthusiasm and helps sales while promoting recycling. Therefore, the online platform is motivated to market and recycle. The advent of blockchain technology will drive the marketing and recycling investments in the platform and improve the consumer experience. Based on the above positive effects brought about by blockchain-enabled recycling, the manufacturer's profits are also improved.

### 5.2. Implementation Conditions for Blockchain-Enabled Recycling

**Theorem 6.** *From the economic benefits of the online platform in different sales models, we deduce the conditions for implementing blockchain-enabled recycling. We identify that the strategic decision to deploy blockchain technology on the online platform depends on the trade-off between the long-term profits and the initial investment cost of blockchain.*

(1) When selling products on a reselling model, the specific condition that the online platform uses blockchain technology to enable recycling is $F < v_1$, where

$$v_1 = \frac{1}{2k_s}\left(\frac{\rho(\theta+\lambda)^2}{16\beta}\right)^2\left[\frac{2r+\delta}{(r+\delta)^2\delta} - \frac{2r+\delta+1-\xi}{(r+\delta+1-\xi)^2(\delta+1-\xi)}\right] + (1-\xi)a\alpha\Delta + \frac{(1-\xi^2)(\alpha\Delta\varepsilon)^2}{2k_u} \quad (12)$$

(2) When selling products on a marketplace model, the specific condition that the online platform uses blockchain technology to enable recycling is $F < v_2$, where

$$v_2 = \frac{1}{2k_s}\left(\frac{\varphi\rho(\theta+\lambda)^2}{4\beta}\right)^2\left(\frac{2r+\delta}{(r+\delta)^2\delta} - \frac{2r+\delta+1-\xi}{(r+\delta+1-\xi)^2(\delta+1-\xi)}\right) + (1-\xi)a\alpha\Delta + \frac{(1-\xi^2)(\alpha\Delta\varepsilon)^2}{2k_u} \quad (13)$$

Theorem 6 illustrates the conditions for the online platform to deploy blockchain technology for recovery. The initial deployment cost of the blockchain must satisfy $V_P^{BR} > V_P^{NR}$ and $V_P^{BM} > V_P^{NM}$ in the reselling and marketplace models, respectively. Intuitively, because of the initial fixed investment cost of blockchain, the online platform will evaluate the profitability of adopting blockchain technology against the initial fixed cost when making a technology adoption decision. Analytically, we find that there is an economic incentive for the online platform to adopt blockchain-enabled recycling. That is, when $F < v_1$ in the reselling mode and $F < v_2$ in the marketplace mode. Since the

complexity of $v_1$ and $v_2$, we will illustrate the specific conditions for blockchain-enabled recycling with the help of numerical examples in Section 6.

*5.3. The Impact of Blockchain on the Optimal Sales Model Selection*

**Theorem 7.** *Interestingly, our research finds that adopting blockchain technology does not impact the optimal sales model selection for the platform and does not change the manufacturer's intention to collaborate. In other words, the sales model preferences of E-CLSC members depend on the commission rate $\varphi$, and this preference will not change with the advent of blockchain technology. Specifically, the platform tends to choose the reselling model when $0 < \varphi \le 1/4$ and the marketplace model when $1/4 < \varphi < 1$, whether the blockchain is implemented or not. Similarly, unaffected by blockchain technology, the manufacturer tends to engage in a reselling model with the online platform if $0 < \varphi < \left(2 - \sqrt{2}\right)/4$ or $\left(2 + \sqrt{2}\right)/4 < \varphi < 1$, and a marketplace model if the commission rate $\varphi$ satisfies $\left(2 - \sqrt{2}\right)/4 < \varphi < \left(2 + \sqrt{2}\right)/4$.*

Theorem 7 shows that using blockchain technology to enable recycling does not change the existing sales model configuration of the enterprises. The existing businesses under each sales model can run based on silent cost inputs such as constructed logistics service systems, warehousing systems, etc. [59]. Therefore, if blockchain technology impacts the sales model in operations, additional costs will be incurred beyond the initial fixed deployment cost, including staff training, seeking new logistics services, etc., which also results in the waste of the established operational system. The protection of existing sales models by blockchain technology can inspire companies to use blockchain technology to address challenges in E-CLSCs, such as opaque recycling chains, consumer mistrust, and difficulties in building corporate image. In addition, we find that the sales model preferences of E-CLSC members are closely linked to the commission rate. If the commission rate is low, the platform chooses the reselling model; if the commission rate is high, the platform chooses the marketplace model. This is consistent with our intuition that a higher commission rate represents a higher marginal profit for the platform when selling a product. The benefits of higher marginal revenue and the avoidance of double marginalization drive the platform to select the marketplace model. In contrast, if the commission rate is lower, the platform earns less as an intermediary, so the reselling model is a better selection. For the manufacturer, we find that the manufacturer prefers the reselling model if the commission rate is extremely low or high, and the marketplace model is a better selection if the commission rate is moderate. This differs from the findings of Xu and Choi [27], in which the manufacturer chose the marketplace model if commission rates are low. In our research, there is an agreeable range between the manufacturer and the platform in terms of commission rate preference. The manufacturer and the platform agree on the marketplace model if the commission rate is moderate and collaborate on the reselling model if the commission rate is extremely low. This indicates that when a manufacturer trusts a platform and entrusts it with marketing and recovery services, the parties can significantly reduce the disagreement over the setting of commission rates. Our analysis will offer actionable insights into the selection of sales models in the E-CLSC context, particularly for enterprises considering equipping with blockchain technology.

## 6. Numerical Study

The comparative analysis in Section 5 reveals that the initial deployment cost of blockchain technology is a crucial factor affecting the adoption of blockchain-enabled recycling, while the commission rate for online sales is an essential factor affecting the sales model selection. The complexity of each scenario and the interactions between critical exogenous parameters cannot be simply represented through comparative study alone. In this section, based on the equilibrium derived in Section 5, we will perform a more intuitive and in-depth analysis with the help of numerical examples. Following the key assumptions in Xu and Choi [64] for blockchain implementation, De Giovanni et al. [75]

for recycling, and Tian et al. [57] for the platform under different sales models, and set to satisfy all model assumptions (positive profits, state, and strategies), the basic parameters are set as follows: $\alpha = 0.2$, $\theta = 3$, $\lambda = 1$, $\beta = 0.4$, $r = 0.1$, $\delta = 0.6$, $\Delta = 1.5$, $\xi = 0.7$, $a = 0.3$, $k_s = 2$, $k_u = 3$, $\varepsilon = 2$, $\rho = 2.5$.

### 6.1. The Impact of Consumer Trust and Marginal Revenue of Used Products on Blockchain-Enabled Recycling

From the above analysis, deploying blockchain technology to empower recycling wins consumers' trust, helps build brand image, and contributes to recycling while increasing product sales. However, we have not touched on the discussion of recycling rates. In the recycling industry, the recycling rate is an important indicator of environmental performance. Some manufacturers are even under legislative pressure to increase recycling rates [76,77]. Therefore, we open the discussion from an environmental perspective to find the feasible region where the advent of blockchain will increase recycling rates, i.e., where environmental performance is improved. We represent the ratio of reverse recycling quantity to forward demand as the recycling rate $\tau$, where $\tau = D_b/D_f$. In the reselling and marketplace modes, environmental performance improvement needs to meet the following conditions: $\tau^{BR} > \tau^{NR}$ and $\tau^{BM} > \tau^{NM}$, respectively. Accordingly, we focus on two important factors in the recycling process, consumer trust $\xi$ and used product marginal revenue $\Delta$, to explore in detail their impact on the deployment of blockchain technology.

Figure 2 illustrates the impact of consumer trust $\xi$ and marginal revenue $\Delta$ of used products on blockchain-enabled recycling. It can be seen that for different used product marginal revenue $\Delta$, blockchain-enabled recycling can improve environmental benefits when the level of consumer trust $\xi$ falls below a certain threshold. This indicates that recycling used products can be carried out effectively when consumers trust the recyclers. At this point, using blockchain technology for recycling makes little sense, and the online platform is more cautious in making a blockchain deployment decision. In addition, when the marginal economic revenue $\Delta$ of the used product is high, recycling using blockchain technology becomes a viable option even if consumer trust $\xi$ is at a high level. This means that items with high residual values are better suited for recycling with blockchain technology, even though consumers may greatly trust the recyclers.

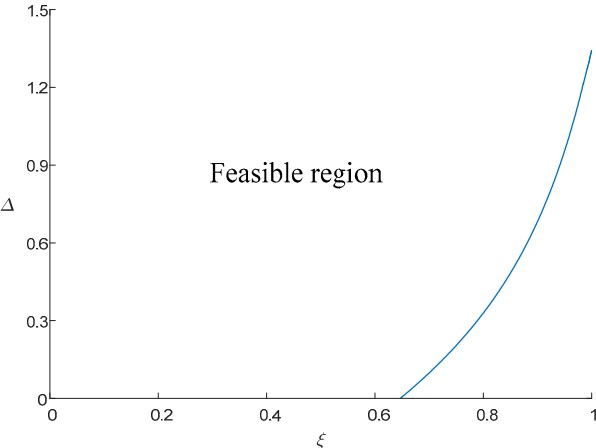

**Figure 2.** The impact of $\xi$ and $\Delta$ on blockchain-enabled recycling.

### 6.2. The Conditions for the Online Platform to Use Blockchain Technology to Enable Recycling and the Sales Model Selection by Each Player

According to the study in Sections 5.2 and 5.3, the initial investment cost $F$ of equipping blockchain technology is an essential factor on whether the online platform decides to adopt blockchain technology to empower recovery, while the optimal configuration of the sales model is linked to the commission rate $\varphi$. Therefore, we consider two critical factors

for the operation of the E-CLSC, initial investment cost $F$ and the commission rate $\varphi$, to explore the adoption of blockchain technology, as shown in Figure 3.

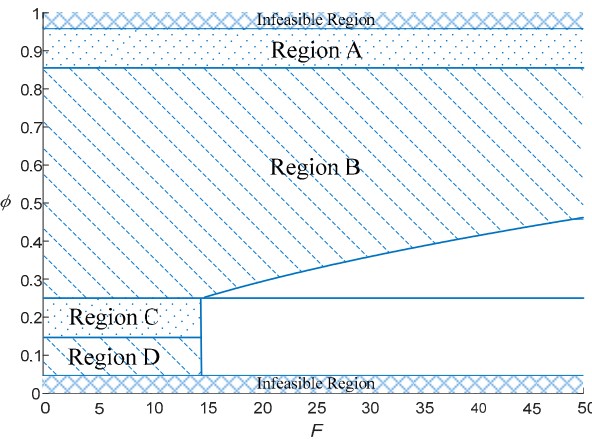

**Figure 3.** The impact of $F$ and $\varphi$ on the adoption of blockchain and sales model selection.

Figure 3 illustrates the effect of the fixed blockchain deployment cost $F$ and the commission rate $\varphi$ on the strategic decision to adopt blockchain and the sales model selection. In the marketplace model, we find that the commission rate $\varphi$ affects the platform's acceptance of blockchain technology. This is because as commission rate $\varphi$ increase, it can help the platform alleviate the financial pressure associated with deploying blockchain technology. Therefore, the acceptance of blockchain technology varies with the sales model. The deployment of blockchain on the platform should be combined with specific sales models.

Based on the positive value function of the online platform and the constraints constructed in the model description, infeasible regions that do not meet the criteria for practical significance are first excluded. From the discussion in Section 5.1, it is clear that the advent of blockchain always benefits the manufacturer. Thus, for the manufacturer, we focus mainly on the consensual region of the sales model choice. For the platform, identify the areas where it strategically decides to deploy blockchain technology under the reselling or marketplace model, with the assurance that implementing blockchain-enabled recycling is profitable. The blank areas in Figure 3 represent areas where the online platform will not decide to deploy blockchain technology because the long-term profitability cannot afford the initial fixed costs of implementing blockchain. For regions $A$, $B$, $C$, and $D$, the specific situations of sales model selection in the case of using blockchain are shown in Table 5.

**Table 5.** Sales model choice when blockchain-enabled recycling.

| | | Region $A$ | | Region $B$ | | Region $C$ | | Region $D$ | |
|---|---|---|---|---|---|---|---|---|---|
| | | **Sales Model Selection for the Online Platform** | | | | | | | |
| **Cooperation Intentions of the Manufacturer** | | Reselling | Marketplace | Reselling | Marketplace | Reselling | Marketplace | Reselling | Marketplace |
| Region $A$ | Reselling | $V_M^R, V_P^R$ | $\underline{V_M^R}, \underline{V_P^M}$ | | | | | | |
| | Marketplace | $V_M^M, V_P^R$ | $V_M^M, V_P^R$ | | | | | | |
| Region $B$ | Reselling | | | $V_M^R, V_P^R$ | $V_M^R, V_P^M$ | | | | |
| | Marketplace | | | $V_M^M, V_P^R$ | $\underline{V_M^M}, \underline{V_P^M}$ | | | | |
| Region $C$ | Reselling | | | | | $V_M^R, V_P^R$ | $V_M^R, V_P^M$ | | |
| | Marketplace | | | | | $\underline{V_M^M}, \underline{V_P^R}$ | $V_M^M, V_P^M$ | | |
| Region $D$ | Reselling | | | | | | | $\underline{V_M^R}, \underline{V_P^R}$ | $V_M^R, V_P^M$ |
| | Marketplace | | | | | | | $V_M^M, V_P^R$ | $V_M^M, V_P^M$ |

Table 5 demonstrates the sales model selection for the platform and the manufacturer's intention to collaborate with it when using blockchain for recycling. For region $A$, where the commission rate of the product is extremely high, the online platform can earn a

considerable marginal return on the sale of a product, so the marketplace model is a better selection. For the manufacturer, the extremely high commission rate forces it to take measures to alleviate the economic disadvantage, such as raising the retail price, etc. However, the high selling price will form resistance to the sales of the product, which still cannot relieve the economic pressure. Therefore, the manufacturer prefers to cooperate in the reselling model to avoid the disadvantages of the marketplace model. In this case, the manufacturer and the platform disagree on the sales model selection and cannot achieve a solid partnership. For region *B*, as the commission rate of the product is at a relatively high level, the platform receives a good income when selling products and therefore has sufficient marketing and recycling funds to promote the orderly conduct of activities in the E-CLSC, so she prefers to select the marketplace model. From the manufacturer's perspective, although the relatively high commission rate gives up some of the revenue, the manufacturer in the marketplace model gains pricing power over retail prices. Under the condition that the platform carries out the sales and recycling activities smoothly, the manufacturer can appropriately increase the retail price to make profits. At this point, even if there are relatively high commission rates, the E-CLSC can still operate stably in the marketplace model. For region *C*, the platform makes less revenue from selling products because commission rates are relatively low, so the reselling model is a better selection. For the manufacturer, the relatively low commission rate improves its profitability in the marketplace model, so the manufacturer prefers the marketplace model. In this situation, E-CLSC members disagree on the selection of the sales model. For region *D*, due to the extremely low commission rates, the online platform does not have sufficient funds to engage in marketing and recycling activities in the marketplace model, which is detrimental to the functioning of the E-CLSC. The inefficient marketplace mode drives the E-CLSC members towards the reselling model. Thus, the platform and the manufacturer have formed a solid partnership under the reselling model.

In short, for region *A* and region *C*, the online platform and the manufacturer disagree on the sales model selection in pursuit of profit maximization. This disagreement is difficult to bridge, even with the enabling effect of blockchain technology. An interesting comparison occurs in region *B* and region *D*, where the online platform and the manufacturer have reached a consensus, the online platform can withstand higher initial deployment costs in region *B*, where the marketplace model is implemented, especially if the products have a high commission rate. This indicates that the elevated commission rates in the marketplace model can boost the confidence of the online platform to use blockchain technology to enable recycling. On this basis, we obtain the areas where blockchain can improve the economic effects, social welfare, and environmental benefits of E-CLSC, as shown in Figure 4.

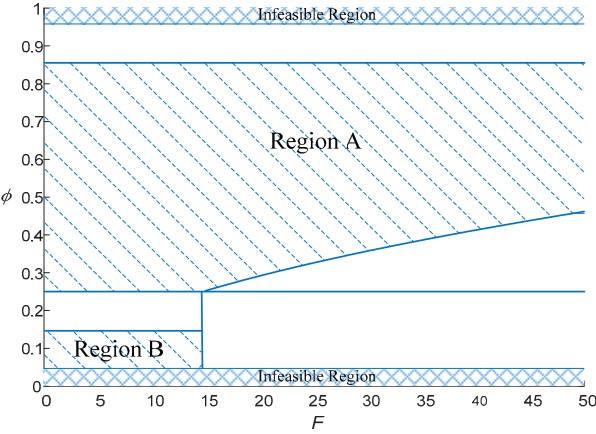

**Figure 4.** Economic effects, social welfare, and environmental benefits of E-CLSC.

Figure 4 illustrates the effects of $F$ and $\varphi$ on the economic effects, social welfare, and environmental benefits of E-CLSC. One of the research objectives of this paper is to find the specific situation where blockchain achieves triple performance improvement. Considering the changes in the consumer surplus and the profits of E-CLSC members and using the increase in recycling rate as the basis for the enhancement of environmental benefits, the regions that enable the improvement of economic effects, social welfare, and environmental benefits in the reselling model or marketplace model are identified. Region $A$ represents the range of the marketplace model chosen by both the manufacturer and the platform, and the E-CLSC achieves triple performance improvement after deploying blockchain technology. Similarly, region $B$ indicates where the E-CLSC achieves triple performance improvement in the reselling model.

### 6.3. The Impact of Consumer Trust on the Performance of the E-CLSC

Subsequently, to explore the impact of other key parameters on the value of blockchain, the initial investment cost of blockchain and the commission rate are set as $F = 10$, $\varphi = 0.3$. We use $E_e$, $E_s$, and $E_g$ to represent economic efficiency, social efficiency, and environmental efficiency, respectively, where $E_e = \frac{(V_M^B + V_P^B) - (V_M^N + V_P^N)}{V_M^N + V_P^N}$, $E_s = \frac{SW^B - SW^N}{SW^N}$, $E_g = \frac{\tau^B - \tau^N}{\tau^N}$.

Table 6 demonstrates the impact of consumer trust $\xi$ on the economic, social, and environmental value of blockchain in the E-CLSC. As we can see, when $\xi$ is at a low level, the use of blockchain technology to enable recycling positively affects economic, social, and environmental efficiency. With an increase in $\xi$, the implementation of blockchain technology gradually adversely affects economic, social, and environmental efficiency. This suggests that when $\xi$ is at a high level, consumers already trust the recyclers, and implementing blockchain technology at this point may harm E-CLSC performance due to the initial investment cost. In addition, blockchain technology can create greater economic, social, and environmental value in less trustworthy business environments when consumers do not trust the recyclers.

**Table 6.** The impact of consumer trust $\xi$ on the triple performance of the E-CLSC.

| | $\xi$ | 0.0 | 0.1 | 0.2 | 0.3 | 0.4 | 0.5 | 0.6 | 0.7 | 0.8 | 0.9 | 1.0 |
|---|---|---|---|---|---|---|---|---|---|---|---|---|
| Reselling | $E_e^R$ | 5.088 | 4.355 | 3.671 | 3.036 | 2.450 | 1.912 | 1.422 | 0.977 | 0.579 | 0.226 | $-0.083$ |
| | $E_s^R$ | 5.152 | 4.414 | 3.725 | 3.085 | 2.493 | 1.950 | 1.454 | 1.004 | 0.601 | 0.243 | $-0.070$ |
| | $E_g^R$ | — | 7.613 | 3.119 | 1.669 | 0.974 | 0.579 | 0.335 | 0.178 | 0.078 | 0.021 | 0 |
| Marketplace | $E_e^M$ | 5.267 | 4.518 | 3.818 | 3.167 | 2.565 | 2.011 | 1.505 | 1.046 | 0.634 | 0.268 | $-0.051$ |
| | $E_s^M$ | 5.344 | 4.590 | 3.885 | 3.228 | 2.619 | 2.059 | 1.546 | 1.081 | 0.663 | 0.292 | $-0.033$ |
| | $E_g^M$ | — | 7.613 | 3.119 | 1.669 | 0.974 | 0.579 | 0.335 | 0.178 | 0.078 | 0.021 | 0 |

Note: — stands for trivial situations.

### 6.4. Time Trajectories of Brand Goodwill and Retail Price

To explore the impact of initial brand goodwill on the equilibria, we set the initial brand goodwill as $G_0 = 0 < G_\infty^i$ and $G_0 = 35 > G_\infty^i$, respectively, where superscript $i \in \{NR, NM, BR, BM\}$, time $t \in [0, 10]$. The specific analysis is as follows.

Figure 5a illustrates the brand goodwill trajectories under four scenarios: $NR$, $NM$, $BR$, $BM$. The brand goodwill reaches steady-state values in all four scenarios, and the equilibria are unaffected by the initial goodwill. The equilibria under the four scenarios satisfy $G_\infty^{BM} > G_\infty^{BR} > G_\infty^{NM} > G_\infty^{NR}$.

Figure 5b illustrates the retail price trajectories under four scenarios: $NR$, $NM$, $BR$, $BM$. The retail prices reach steady-state values in all four scenarios, and the equilibria are unaffected by the initial goodwill. The equilibria under the four scenarios satisfy $p_\infty^{BR} > p_\infty^{BM} > p_\infty^{NR} > p_\infty^{NM}$.

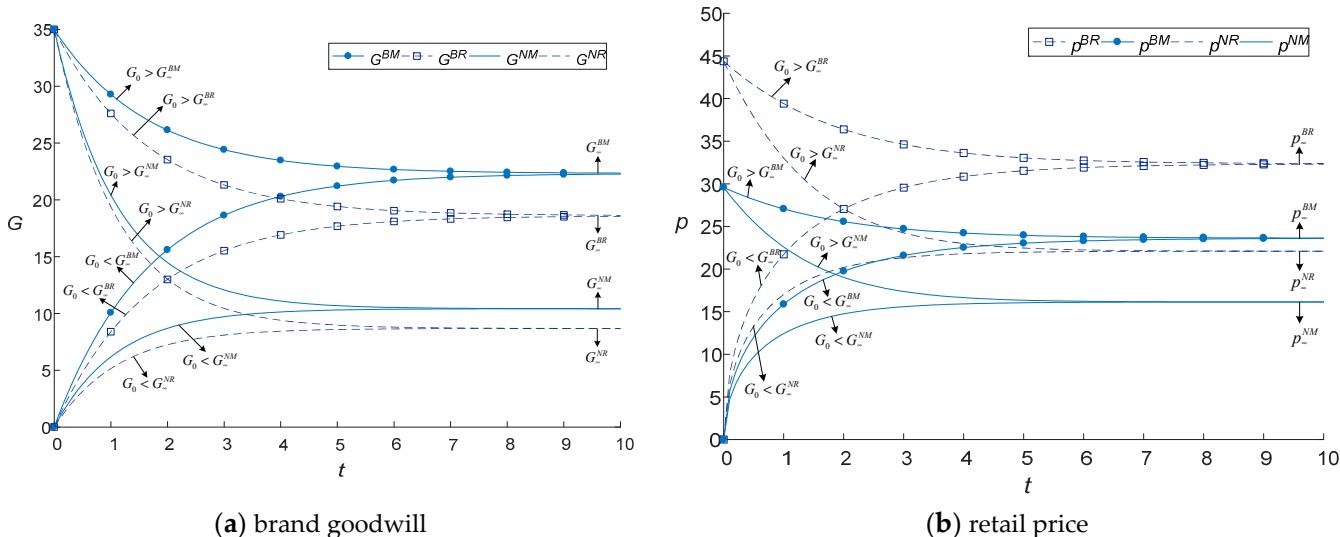

(**a**) brand goodwill

(**b**) retail price

**Figure 5.** Brand goodwill and retail price trajectories with different initial goodwill.

When the online platform invests in blockchain to enable recycling, the transparent visualization of the used product recycling process wins consumer trust and helps build the corporate image. In this way, consumers are more willing to take action on recycling rather than wait and see. The establishment of the corporate image benefits from blockchain technology. More specifically, the recycling side carries blockchain for recycling, which can be regarded as a positive marketing means. To some extent, it increases consumers' favorability towards the enterprise and drives product sales. Therefore, the manufacturer or platform can increase the products' retail price appropriately. As a result, brand goodwill and retail prices are increased when blockchain technology is deployed to enable recycling.

### 6.5. The Impact of Consumer Trust on Brand Goodwill, Manufacturer Profits, Online Platform Profits, and Social Welfare

The implementation of blockchain-enabled recycling will differ in different trust environments. In this subsection, we explore the impact of consumer trust on brand goodwill, manufacturer profits, online platform profits, and social welfare. The specific analysis is as follows.

Figure 6 illustrates the impact of consumer trust $\zeta$ on brand goodwill, E-CLSC members' profits, and social welfare. We find that blockchain-enabled recycling can effectively improve brand goodwill and manufacturer profits in reselling and marketplace models. Therefore, the manufacturer always supports adopting blockchain technology to enable recycling in order to maintain a positive corporate image and profitability. As consumer trust $\zeta$ increases, the value created by blockchain decreases. When the consumer trust level $\zeta$ is low, implementing blockchain technology can effectively tackle the issues of unclear recycling processes and consumer mistrust, which benefits recycling and corporate image building. At this time, blockchain adoption can enhance online platform's profits. The lower the level of consumer trust $\zeta$, the better the improvement in the profitability of the online platform enabled by blockchain technology. When consumer trust $\zeta$ is high, the value created by blockchain technology is lower than the financial pressure placed on the online platform. Therefore, it is more profitable for the platform not to consider blockchain technology. The impact of $\zeta$ on social welfare is the same.

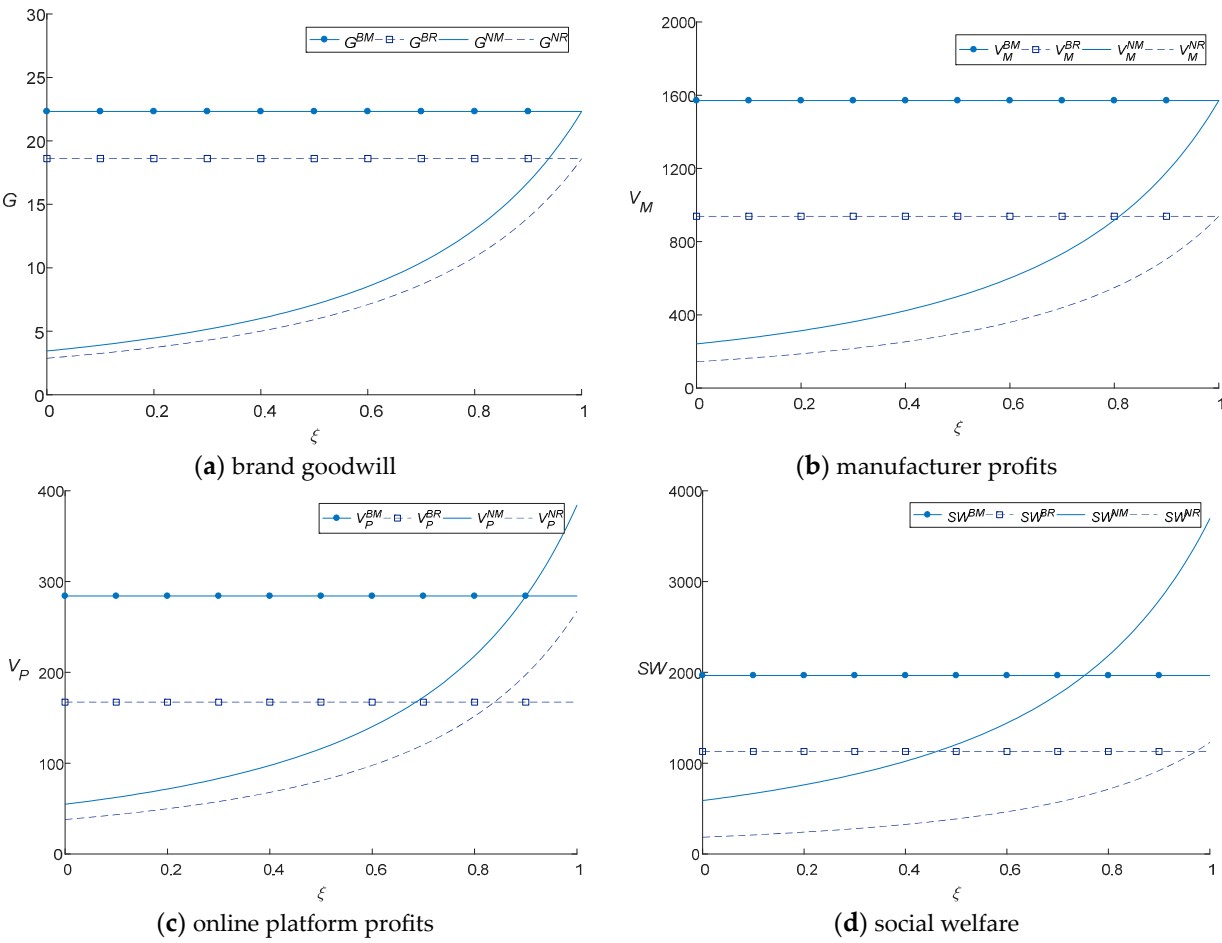

(**a**) brand goodwill

(**b**) manufacturer profits

(**c**) online platform profits

(**d**) social welfare

**Figure 6.** The impact of consumer trust $\xi$ on brand goodwill, manufacturer profits, online platform profits, and social welfare.

### 6.6. The Impact of Platform Power on Brand Goodwill, Manufacturer Profits, Online Platform Profits, and Social Welfare

In practice, each platform's key resources and core capabilities will vary. Therefore, in this subsection, we investigate the improvement effect of blockchain technology under different platform power as follows.

Figure 7 illustrates the impact of platform power $\lambda$ on brand goodwill, E-CLSC members' profits, and social welfare. As the platform power $\lambda$ grows, the platform has more key resources and stronger core capabilities, such as more user resources, stronger technical and business capabilities, etc. This makes it easier for the manufacturer to gain competitive advantages and facilitates the various activities in the E-CLSC. Thus, brand goodwill, E-CLSC members' profits, and social welfare are enhanced. Increasing platform power is always advantageous, so online platforms are willing to put in efforts to grow their platform power, and manufacturers prefer to join online platforms that have strong platform power. In addition, comparing the same sales model with and without blockchain shows that the stronger the platform power, the better blockchain improves. This suggests that blockchain creates greater value in business environments with greater platform power and implies that online platforms with more key resources and stronger core capabilities hold greater incentives to deploy blockchain.

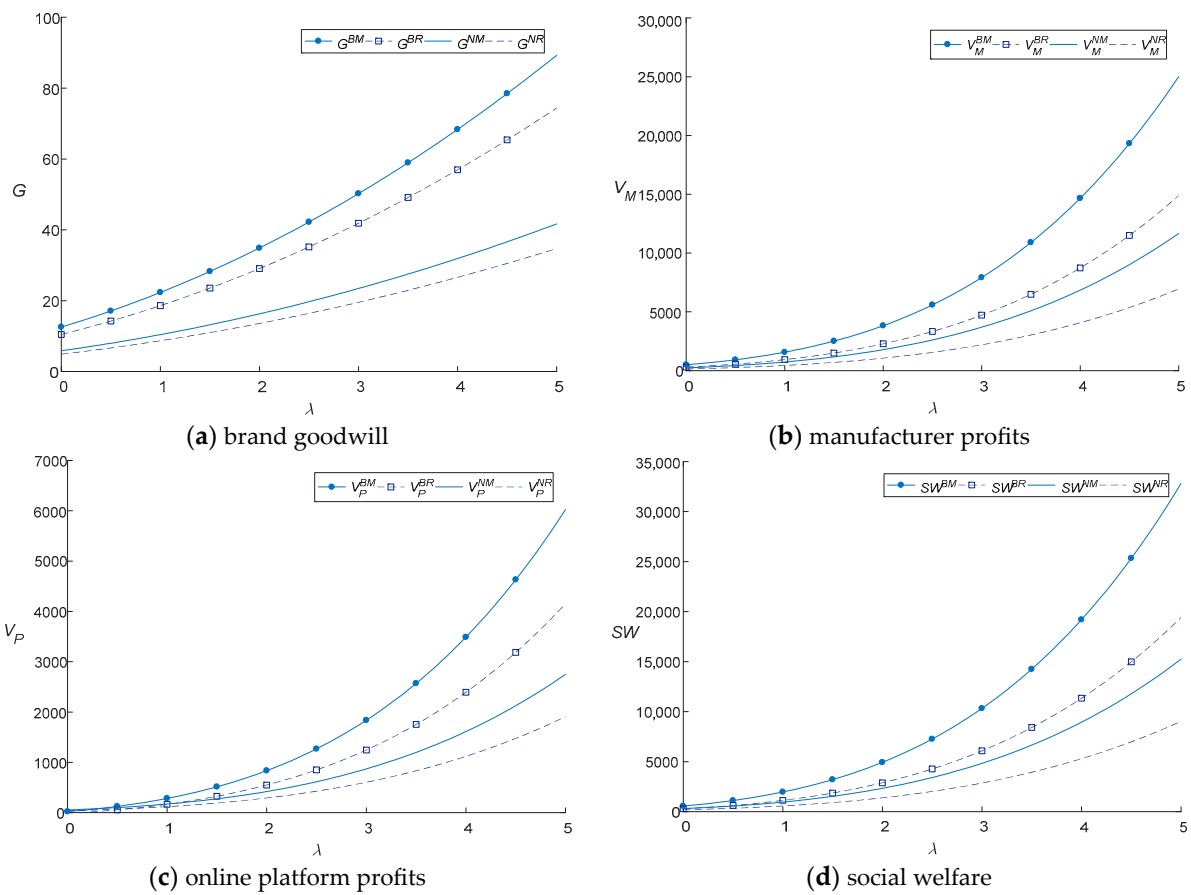

**Figure 7.** The impact of platform power $\lambda$ on brand goodwill, manufacturer profits, online platform profits, and social welfare.

## 7. Extension: Per-Unit Blockchain Cost

In practice, it is a common approach to partner with blockchain technology providers in order to use blockchain technology to address challenges. For example, Walmart and Nestle have undertaken major blockchain collaborations with IBM in order to improve supply chain traceability [78,79]. Everledger provides blockchain technology support to diamond companies working to address diamond authentication and certification problems [45]. In the recycling industry, there are also many blockchain technology providers. For example, RecycleGo in the United States offers blockchain-based recycling for commercial waste, and Agora Tech Lab in the Netherlands uses blockchain technology for recycling household and community waste [11]. Therefore, when the profitability of the online platform is not enough to support building a blockchain technology-enabled recycling system, it is more likely that she will start a partnership with a blockchain technology provider. At this point, we assume the platform does not need to bear the blockchain deployment cost but instead incurs one unit of blockchain cost $b$ in each used product tracking process [74]. To demonstrate the robustness of our findings more visually, this section examines the impact of per-unit blockchain cost $b$ and commission rate $\varphi$ on the sales model choice and the E-CLSC triple performance.

**Theorem 8.** *The equilibrium strategies for E-CLSC members with per-unit blockchain cost; the detailed analysis is given in the Appendix A.*

In scenario *BR*, the optimization problem can be summarized as

$$\max_{w}\left\{J_M^{BR} = \int_0^\infty e^{-rt}\left[w(t)D_f^{BR} + (1-\alpha)(\Delta-b)D_b^{BR}\right]dt\right\}$$

$$s.t.\begin{cases} \max_{p,u,s}\left\{J_P^{BR} = \int_0^\infty e^{-rt}\left[(p(t)-w(t))D_f^{BR} + \alpha(\Delta-b)D_b^{BR} - \frac{1}{2}k_u u(t)^2 - \frac{1}{2}k_s s(t)^2\right]dt\right\} \\ \dot{G}(t) = \rho s(t) - \delta G(t) \quad G(0) = G_0 \end{cases} \tag{14}$$

In scenario *BM*, the optimization problem can be summarized as

$$\max_{p}\left\{J_M^{BM} = \int_0^\infty e^{-rt}\left[(1-\varphi)p(t)D_f^{BM} + (1-\alpha)(\Delta-b)D_b^{BM}\right]dt\right\}$$

$$s.t.\begin{cases} \max_{u,s}\left\{J_P^{BM} = \int_0^\infty e^{-rt}\left[\varphi p(t)D_f^{BM} + \alpha(\Delta-b)D_b^{BM} - \frac{1}{2}k_u u(t)^2 - \frac{1}{2}k_s s(t)^2\right]dt\right\} \\ \dot{G}(t) = \rho s(t) - \delta G(t) \quad G(0) = G_0 \end{cases} \tag{15}$$

Figure 8 shows the impact of commission rate $\varphi$ and per-unit blockchain cost $b$ on the blockchain deployment decision and the sales model selection. Compared to the case of fixed blockchain deployment costs, the distinction is that when the per-unit blockchain cost falls below a specific threshold, blockchain technology benefits both the reselling and marketplace models. The blank areas in Figure 8 represent the cases where blockchain is not used or where it does not make sense. Consistent with the findings in Section 6.2, blockchain-enabled recycling at a unit cost does not affect the sales model selection. The optimal sales model configuration is only linked to commission rate $\varphi$.

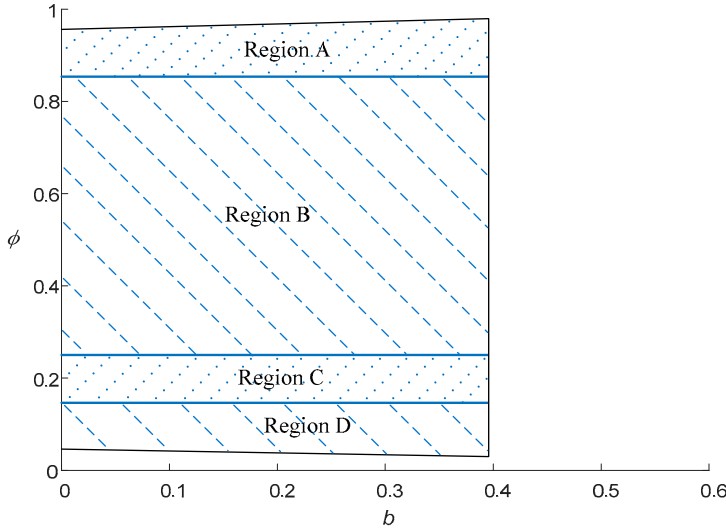

**Figure 8.** The impact of commission rate $\varphi$ and per-unit blockchain cost $b$ on the adoption of blockchain technology and the sales model selection.

Figure 9 depicts the influence of per-unit blockchain cost $b$ and commission rate $\varphi$ on the economic benefits, social welfare, and environmental benefits of the E-CLSC when the platform partners with a blockchain provider. Regions $A$ and $B$ represent areas for improving E-CLSC performance under the marketplace and reselling models, respectively. We find that the triple performance improvement of the E-CLSC from blockchain-enabled recycling remains even as the blockchain usage cost moves from an initial deployment cost to a per-unit cost. Accordingly, the core conclusion of our research, namely the implementation of blockchain technology does not affect the optimal configuration of sales model selection and that blockchain-enabled recycling improves the E-CLSC performance, remains robust in the extended model. In summary, to take advantage of blockchain-enabled recycling, platforms can either cooperate with a blockchain technology provider or establish their own blockchain. There is little difference between the two.

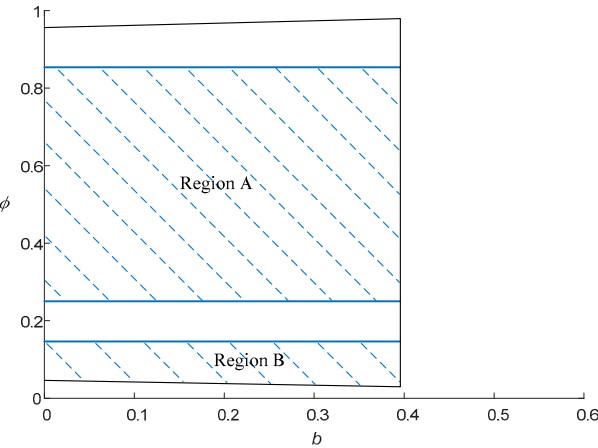

**Figure 9.** Economic effects, social welfare, and environmental benefits of E-CLSC.

## 8. Conclusions and Management Insights

### 8.1. Concluding Remarks

To explore the value of blockchain technology in addressing the lack of consumer trust in recyclers, we analyze an E-CLSC composed of an online platform and a manufacturer, where the online platform is responsible for product marketing and used product recycling. When blockchain is not used, there is no reliable proof of the sustainable behavior of recycling chain members, and consumers' mistrust of recyclers can affect the realization of recycling actions and discourage brand goodwill. And when using blockchain technology to enable recycling, consumers can view the disposal process of used products, solving the trust problem and providing a foundation for improving the efficiency of the E-CLSC. We characterize the equilibrium strategies for the online platform and the manufacturer. Our analysis reveals the following:

First, by comparing the equilibrium results with and without blockchain-enabled recycling, we summarize the influence of blockchain technology on the optimal decision-making. Specifically, adopting blockchain technology can effectively enhance the brand goodwill and retail price, contribute to platform's marketing and recycling efforts, and increase long-term consumer demand and the used product recycling quantity. The platform using blockchain for recycling is always beneficial for the manufacturer because the advent of blockchain technology engenders consumer trust and makes the whole E-CLSC more efficient, thus increasing the manufacturer's profits. In different sales models, the platform tends to adopt blockchain if the long-term profits outweigh the initial deployment costs. Considering the case of the online platform working with a blockchain technology provider, the cost of blockchain implementation does not affect our conclusions when it changes from a fixed cost to a per-unit usage cost.

Second, we investigate whether the blockchain implementation will affect the sales model selection. Interestingly, blockchain technology can be adapted to the existing sales model. The optimal sales model selection for the platform and the cooperation intentions of the manufacturer do not change with the advent of blockchain technology. Our study finds that the optimal configuration of the sales model is linked to the commission rate. If commission rates are relatively high (extremely low), the platform and the manufacturer form a solid partnership in the marketplace (reselling) model. However, in other cases, they disagree on the selection of the sales model. In addition, we indicate that the commission rate size will influence the online platform's blockchain deployment decision. As commission rates increase, the online platform can afford higher initial blockchain deployment costs in the marketplace model.

Finally, the value of blockchain-enabled recycling is measured by examining the changes in the economic, social, and environmental performance of the E-CLSC. We reveal that the value of blockchain-enabled recycling is influenced by consumer trust and that blockchain will create more value at lower levels of consumer trust. In detail, implementing

blockchain technology improves the E-CLSC performance when the level of consumer trust falls below a specific threshold. When consumer trust is high, blockchain technology may even harm the "triple performance" of the E-CLSC. For different types of used products, it is found that those with high residual values are more suitable for recycling with blockchain technology. The value of blockchain is also affected by platform power. The stronger the platform power, the greater the value blockchain technology creates.

*8.2. Managerial Implications*

Based on the research results, we provide tangible insights for related practitioners and conclude this paper by discussing the following three aspects.

(1) Companies can adopt blockchain technology to engender consumer trust because the visualization of the recycling process helps companies build a good image, promoting recycling while facilitating sales. In terms of recycling used products, online platforms should increase their recycling efforts to meet the consumer recycling enthusiasm ignited by blockchain technology. For example, improving the door-to-door recycling service process, enhancing staff training, and upgrading big data service technology will help more accurately and comprehensively meet the surging consumer recycling requests. In the process of product sales, the shaping of a corporate image becomes more efficient. Platforms should take this opportunity to increase marketing investment and further enhance the stimulating effect of goodwill on demand. The market competitiveness of products then increases, and companies can raise product prices accordingly to make profits. Specifically, in the reselling model, manufacturers can raise wholesale prices, and online platforms can sell products at higher retail prices; in the marketplace model, manufacturers can set higher retail prices, and online platforms profits from higher retail prices with commission rates. In the initial stage of blockchain implementation, which is economically difficult, manufacturers and platforms should be patient and wait for time, perform their respective functions well, and rely on the enabling role of blockchain technology to make profits in the long run and promote sustainable development. (2) The use of blockchain technology depends on a case-by-case basis. Partnering with blockchain technology providers is also a good option when online platforms are not profitable enough to support the initial deployment costs of blockchain technology. The level of consumer trust affects the value of blockchain technology; the lower the level of consumer trust, the better the improvement of blockchain technology. Therefore, enterprises that have already triggered a consumer trust crisis are better suited to piggyback on blockchain technology. In contrast, those that have already established a trusting relationship with consumers should be cautious about blockchain. For different sales modes, online platforms can withstand higher initial deployment costs in the marketplace mode. The specific acceptance degree depends on the commission rate of the product. In particular, the higher the commission rate, the greater the acceptance degree of the blockchain investment cost. It suggests that marketplace-based online platforms such as Tmall.com and Taobao.com are better suited to carry blockchain technology. For different types of used products, the higher the remanufacturing revenue of the used product, the more suitable it is for recycling using blockchain technology. This encourages companies with the higher residual value of the used product to participate in blockchain recycling, such as Hewlett-Packard, Apple, etc. In addition, the stronger the platform power, the greater the value blockchain creates. Online platforms with more core resources and stronger business capabilities should actively deploy blockchain technology. And platforms that have already deployed blockchain technology should take steps to further enhance their platform power to gain greater blockchain-enabled value. Manufacturers should look for online platforms with stronger platform power to start cooperating to expand the market scale and gain a competitive advantage. (3) The advent of blockchain technology does not impact the existing ecology of the sales model. This has encouraged online platforms that have formed stable partnerships with manufacturers to further consider using blockchain technology to enable recycling, as the advent of blockchain technology does not incur the cost of operational structure

changes. For manufacturers and platforms that have not yet entered a partnership, we recommend a sales partnership at different commission rates based on product type. For example, for products that charge extremely low commission rates, such as auto parts and digital products, cooperation should be carried out under the reselling model, and the deployment of blockchain technology should be further considered. On the other hand, for some products that charge relatively high commission rates, such as artwork, clothing, and accessories, cooperation should be carried out under the marketplace model, and the implementation of blockchain technology should be decided according to specific circumstances.

*8.3. Limitations and Future Research*

In view of the limitations of this paper, several directions are provided for future research. This paper only focuses on online recycling, an emerging recycling method. Indeed, other recycling methods are widely available in practice. For example, in the cemented carbide industry, Sandvik is committed to recycling used products from customers to obtain key raw materials for the products [80]. Microsoft outsources the collection activities of used products to third-party companies [81]. It will be interesting to consider the enabling value of blockchain technology under multiple recycling channels in future research. Second, we assume that using blockchain technology can help win consumer trust. Blockchain technology can be an effective "trust machine" to enable consumer recycling behavior. However, in practice, consumer acceptance of blockchain technology may vary. Future research could explore consumer heterogeneity, where some consumers trust blockchain technology, but others may not be influenced by it. Finally, we only consider the online platform operating under a reselling or marketplace model. Future research could consider hybrid models [57]. It would be interesting to explore the influence of blockchain on multiple sales models and examine the enabling value of blockchain technology at that time.

**Author Contributions:** Formal analysis, D.M.; methodology, D.M., P.M. and J.H.; supervision, P.M. and J.H.; writing—original draft, D.M.; writing—review and editing, P.M. and J.H. All authors have read and agreed to the published version of the manuscript.

**Funding:** This research was funded by the National Natural Science Foundation of China (72202113) and Natural Science Foundation of Shandong Province (ZR2022QG017; ZR2023MG063).

**Data Availability Statement:** Formal dataset was not used in this research. All the data were used in this research are displayed in Appendix A.

**Conflicts of Interest:** The authors declare that they have no known competing financial interests or personal relationships that could have appeared to influence the work reported in this paper.

**Appendix A**

**Proof of Theorem 1.** In scenario $NR$, we need to establish the existence of continuously differentiable value functions $V_M^{NR}$ and $V_P^{NR}$ to ensure that there is a unique solution $G(t)$ to Equation (3) and the $HJB$ equations. The $HJB$ equations for each player are as follows:

$$rV_M^{NR} = \max_{w}\left\{ wD_f^{NR} + (1-\alpha)\Delta D_b^{NR} + \frac{\partial V_M^{NR}}{\partial G}(\rho s - (\delta + 1 - \xi)G) \right\} \tag{A1}$$

$$rV_P^{NR} = \max_{p,u,s}\left\{ (p-w)D_f^{NR} + \alpha \Delta D_b^{NR} - \frac{1}{2}k_u u^2 - \frac{1}{2}k_s s^2 + \frac{\partial V_P^{NR}}{\partial G}(\rho s - (\delta + 1 - \xi)G) \right\} \tag{A2}$$

Since the game is a Stackelberg game and $M$ is the leader, we first determine the first-order necessary conditions for scenario $NR$ as $p^{NR} = \frac{(\theta+\lambda)\sqrt{G}+\beta w}{2\beta}$, $s^{NR} = \frac{\rho}{k_s}\frac{\partial V_P^{NR}}{\partial G}$,

and $u^{NR} = \frac{\alpha\Delta\xi\varepsilon}{k_u}$. Substituting these into $M$'s $HJB$ Equation (A1), the following equation can be obtained:

$$rV_M^{NR} = \max_w \left\{ \frac{(\theta+\lambda)w\sqrt{G} - \beta w^2}{2} + (1-\alpha)\Delta\xi\left(a + \frac{\alpha\Delta\xi\varepsilon^2}{k_u}\right) + \frac{\partial V_M^{NR}}{\partial G}\left(\frac{\rho^2}{k_s}\frac{\partial V_P^{NR}}{\partial G} - (\delta+1-\xi)G\right) \right\} \quad \text{(A3)}$$

Thus, the necessary condition for $M$ is $w^{NR} = \frac{(\theta+\lambda)\sqrt{G}}{2\beta}$. And we obtain the optimal decisions of the players to the game as Equation (A4). □

$$\begin{cases} w^{NR} = \frac{(\theta+\lambda)\sqrt{G}}{2\beta} \\ p^{NR} = \frac{3(\theta+\lambda)\sqrt{G}}{4\beta} \\ s^{NR} = \frac{\rho}{k_s}\frac{\partial V_P^{NR}}{\partial G} \\ u^{NR} = \frac{\alpha\Delta\xi\varepsilon}{k_u} \end{cases} \quad \text{(A4)}$$

Replacing the strategies of each player in Equations (A1) and (A2), we obtain Equations (A5) and (A6):

$$rV_M^{NR} = \frac{(\theta+\lambda)^2}{8\beta}G + (1-\alpha)\Delta\xi\left(a + \frac{\alpha\Delta\xi\varepsilon^2}{k_u}\right) + \frac{\partial V_M^{NR}}{\partial G}\left(\frac{\rho^2}{k_s}\frac{\partial V_P^{NR}}{\partial G} - (\delta+1-\xi)G\right) \quad \text{(A5)}$$

$$rV_P^{NR} = \frac{(\theta+\lambda)^2}{16\beta}G + \alpha a\Delta\xi + \frac{(\alpha\Delta\xi\varepsilon)^2}{2k_u} + \frac{\rho^2}{2k_s}\left(\frac{\partial V_P^{NR}}{\partial G}\right)^2 - (\delta+1-\xi)G\frac{\partial V_P^{NR}}{\partial G} \quad \text{(A6)}$$

Based on the structure of (A5) and (A6), we conjecture the linear-valued functions, $V_M^{NR} = l_1^{NR}G + l_2^{NR}$ and $V_P^{NR} = l_3^{NR}G + l_4^{NR}$, where $l_1^{NR}$, $l_2^{NR}$, $l_3^{NR}$, and $l_4^{NR}$ are the constant parameters to be identified. Substituting $V_M^{NR}$ and $V_P^{NR}$ and their derivatives into Equations (A5) and (A6), we obtain Equations (A7) and (A8):

$$r\left(l_1^{NR}G + l_2^{NR}\right) = \left(\frac{(\theta+\lambda)^2}{8\beta} - (\delta+1-\xi)l_1^{NR}\right)G + (1-\alpha)a\Delta\xi + \frac{(1-\alpha)\alpha(\Delta\xi\varepsilon)^2}{k_u} + \frac{\rho^2}{k_s}l_1^{NR}l_3^{NR} \quad \text{(A7)}$$

$$r\left(l_3^{NR}G + l_4^{NR}\right) = \left(\frac{(\theta+\lambda)^2}{16\beta} - (\delta+1-\xi)l_3^{NR}\right)G + \alpha a\Delta\xi + \frac{(\alpha\Delta\xi\varepsilon)^2}{2k_u} + \frac{\rho^2}{2k_s}\left(l_3^{NR}\right)^2 \quad \text{(A8)}$$

then we can identify the parameter values as follows:

$$\begin{cases} l_1^{NR} = \frac{(\theta+\lambda)^2}{8\beta(r+\delta+1-\xi)} \\ l_2^{NR} = \frac{1}{r}\left[(1-\alpha)a\Delta\xi + \frac{(1-\alpha)\alpha(\Delta\xi\varepsilon)^2}{k_u} + \frac{\rho^2}{2k_s}\left(\frac{(\theta+\lambda)^2}{8\beta(r+\delta+1-\xi)}\right)^2\right] \\ l_3^{NR} = \frac{(\theta+\lambda)^2}{16\beta(r+\delta+1-\xi)} \\ l_4^{NR} = \frac{1}{r}\left[a\alpha\Delta\xi + \frac{(\alpha\Delta\xi\varepsilon)^2}{2k_u} + \frac{\rho^2}{2k_s}\left(\frac{(\theta+\lambda)^2}{16\beta(r+\delta+1-\xi)}\right)^2\right] \end{cases} \quad \text{(A9)}$$

which are all strictly positive; therefore, the results show the concave profit functions with respect to the players' decision variables and the existence of a unique equilibrium that maximizes the objective function. The optimal value functions then become

$$V_M^{NR} = \frac{(\theta+\lambda)^2 G}{8\beta(r+\delta+1-\xi)} + \frac{1}{r}\left[(1-\alpha)a\Delta\xi + \frac{(1-\alpha)\alpha(\Delta\xi\varepsilon)^2}{k_u} + \frac{\rho^2}{2k_s}\left(\frac{(\theta+\lambda)^2}{8\beta(r+\delta+1-\xi)}\right)^2\right] \quad \text{(A10)}$$

$$V_P^{NR} = \frac{(\theta + \lambda)^2 G}{16\beta(r + \delta + 1 - \xi)} + \frac{1}{r}\left[a\alpha\Delta\xi + \frac{(\alpha\Delta\xi\varepsilon)^2}{2k_u} + \frac{\rho^2}{2k_s}\left(\frac{(\theta + \lambda)^2}{16\beta(r + \delta + 1 - \xi)}\right)^2\right] \quad (A11)$$

By solving the differential equation of Equation (3), we obtain the optimal brand goodwill trajectory as $G^{NR} = \left(G_0 - G_\infty^{NR}\right)e^{-(\delta + 1 - \xi)t} + G_\infty^{NR}$, where $G_\infty^{NR} = \frac{\rho^2(\theta + \lambda)^2}{16\beta k_s(r + \delta + 1 - \xi)(\delta + 1 - \xi)}$, $G_\infty^{NR}$ is the steady-state goodwill.

Substituting $G_\infty^{NR}$ into Equations (A10) and (A11), E-CLSC members profits are solved; replacing $G_\infty^{NR}$ and Equation (A9) into Equation (A4), E-CLSC members' equilibrium strategies are solved. Theorem 1 is proved.

**Proof of Theorem 2.** In scenario $NM$, we need to establish the existence of continuously differentiable value functions $V_M^{NM}$ and $V_P^{NM}$ to ensure that there is a unique solution $G(t)$ to Equation (3) and the $HJB$ equations. The $HJB$ equations for each player are as follows:

$$rV_M^{NM} = \max_p\left\{(1 - \varphi)pD_f^{NM} + (1 - \alpha)\Delta D_b^{NM} + \frac{\partial V_M^{NM}}{\partial G}(\rho s - (\delta + 1 - \xi)G)\right\} \quad (A12)$$

$$rV_P^{NM} = \max_{u,s}\left\{\varphi p(t)D_f^{NM} + \alpha\Delta D_b^{NM} - \frac{1}{2}k_u u^2 - \frac{1}{2}k_s s^2 + \frac{\partial V_P^{NM}}{\partial G}(\rho s - (\delta + 1 - \xi)G)\right\} \quad (A13)$$

Since $M$ is the leader, we first determine the first-order necessary conditions for scenario $NM$ as $s^{NM} = \frac{\rho}{k_s}\frac{\partial V_P^{NM}}{\partial G}$ and $u^{NM} = \frac{\alpha\Delta\xi\varepsilon}{k_u}$. Substituting these into $M$'s $HJB$ Equation (A12), the following equation can be obtained:

$$rV_M^{NM} = \max_p\left\{\begin{array}{l}(1 - \varphi)p\left((\theta + \lambda)\sqrt{G} - \beta p\right) \\ +(1 - \alpha)\Delta\xi\left(a + \frac{\alpha\Delta\xi\varepsilon^2}{k_u}\right) + \frac{\partial V_M^{NM}}{\partial G}\left(\frac{\rho^2}{k_s}\frac{\partial V_P^{NM}}{\partial G} - (\delta + 1 - \xi)G\right)\end{array}\right\} \quad (A14)$$

Thus, the necessary condition for $M$ is $p^{NM} = \frac{(\theta + \lambda)\sqrt{G}}{2\beta}$. And we obtain the optimal decisions of the players to the game as Equation (A15). □

$$\begin{cases} p^{NM} = \frac{(\theta + \lambda)\sqrt{G}}{2\beta} \\ s^{NM} = \frac{\rho}{k_s}\frac{\partial V_P^{NM}}{\partial G} \\ u^{NM} = \frac{\alpha\Delta\xi\varepsilon}{k_u} \end{cases} \quad (A15)$$

Replacing the strategies of each player in Equations (A12) and (A13), we obtain the following:

$$rV_M^{NM} = \frac{(1 - \varphi)(\theta + \lambda)^2}{4\beta}G + (1 - \alpha)\Delta\xi\left(a + \frac{\alpha\Delta\xi\varepsilon^2}{k_u}\right) + \frac{\partial V_M^{NM}}{\partial G}\left(\frac{\rho^2}{k_s}\frac{\partial V_P^{NM}}{\partial G} - (\delta + 1 - \xi)G\right) \quad (A16)$$

$$rV_P^{NM} = \left(\frac{\varphi(\theta + \lambda)^2}{4\beta}\right)G + \alpha a\Delta\xi + \frac{(\alpha\Delta\xi\varepsilon)^2}{2k_u} + \frac{\rho^2}{2k_s}\left(\frac{\partial V_P^{NM}}{\partial G}\right)^2 - (\delta + 1 - \xi)G\frac{\partial V_P^{NM}}{\partial G} \quad (A17)$$

Based on the structure of (A16) and (A17), we conjecture the linear-valued functions, $V_M^{NM} = l_1^{NM}G + l_2^{NM}$ and $V_P^{NM} = l_3^{NM}G + l_4^{NM}$, where $l_1^{NM}, l_2^{NM}, l_3^{NM}$, and $l_4^{NM}$ are the constant parameters to be identified. Substituting $V_M^{NM}$ and $V_P^{NM}$ and their derivatives into Equations (A16) and (A17) we obtain the following:

$$r\left(l_1^{NM}G + l_2^{NM}\right) = \begin{cases}\left(\frac{(1 - \varphi)(\theta + \lambda)^2}{4\beta} - (\delta + 1 - \xi)l_1^{NM}\right)G \\ +(1 - \alpha)a\Delta\xi + \frac{(1 - \alpha)\alpha(\Delta\xi\varepsilon)^2}{k_u} + \frac{\rho^2}{k_s}l_1^{NM}l_3^{NM}\end{cases} \quad (A18)$$

$$r\left(l_3^{NM}G + l_4^{NM}\right) = \left(\frac{\varphi(\theta+\lambda)^2}{4\beta} - (\delta+1-\xi)l_3^{NM}\right)G + \alpha a\Delta\xi + \frac{(\alpha\Delta\xi\varepsilon)^2}{2k_u} + \frac{\rho^2}{2k_s}\left(l_3^{NM}\right)^2 \tag{A19}$$

then we can identify the parameter values as follows:

$$\begin{cases} l_1^{NM} = \frac{(1-\varphi)(\theta+\lambda)^2}{4\beta(r+\delta+1-\xi)} \\ l_2^{NM} = \frac{1}{r}\left[(1-\alpha)\Delta\xi a + \frac{(1-\alpha)\alpha(\Delta\varepsilon\xi)^2}{k_u} + \frac{\rho^2(1-\varphi)\varphi}{k_s}\left(\frac{(\theta+\lambda)^2}{4\beta(r+\delta+1-\xi)}\right)^2\right] \\ l_3^{NM} = \frac{\varphi(\theta+\lambda)^2}{4\beta(r+\delta+1-\xi)} \\ l_4^{NM} = \frac{1}{r}\left[a\alpha\Delta\xi + \frac{(\alpha\Delta\varepsilon\xi)^2}{2k_u} + \frac{\rho^2}{2k_s}\left(\frac{\varphi(\theta+\lambda)^2}{4\beta(r+\delta+1-\xi)}\right)^2\right] \end{cases} \tag{A20}$$

which are all strictly positive; therefore, the results show the concave profit functions with respect to the players' decision variables and the existence of a unique equilibrium that maximizes the objective function. The optimal value functions then become

$$V_M^{NM} = \frac{(1-\varphi)(\theta+\lambda)^2}{4\beta(r+\delta+1-\xi)}G + \frac{1}{r}\left[\begin{array}{c}(1-\alpha)\Delta\xi a \\ + \frac{(1-\alpha)\alpha(\Delta\varepsilon\xi)^2}{k_u} + \frac{\rho^2(1-\varphi)\varphi}{k_s}\left(\frac{(\theta+\lambda)^2}{4\beta(r+\delta+1-\xi)}\right)^2\end{array}\right] \tag{A21}$$

$$V_P^{NM} = \frac{\varphi(\theta+\lambda)^2}{4\beta(r+\delta+1-\xi)}G + \frac{1}{r}\left[a\alpha\Delta\xi + \frac{(\alpha\Delta\varepsilon\xi)^2}{2k_u} + \frac{\rho^2}{2k_s}\left(\frac{\varphi(\theta+\lambda)^2}{4\beta(r+\delta+1-\xi)}\right)^2\right] \tag{A22}$$

By solving the differential equation of Equation (3), we obtain the optimal brand goodwill trajectory as $G^{NM} = \left(G_0 - G_\infty^{NM}\right)e^{-(\delta+1-\xi)t} + G_\infty^{NM}$, where $G_\infty^{NM} = \frac{\varphi\rho^2(\theta+\lambda)^2}{4\beta k_s(r+\delta+1-\xi)(\delta+1-\xi)}$, $G_\infty^{NM}$ is the steady-state goodwill.

Substituting $G_\infty^{NM}$ into Equations (A21) and (A22), E-CLSC members profits are solved; replacing $G_\infty^{NM}$ and Equation (A20) into (A15), E-CLSC members' equilibrium strategies are solved. Theorem 2 is proved.

**Proof of Corollary 1.** In scenarios *NR* and *NM*, the platform profits are $V_P^{NR} = l_3^{NR}G_\infty^{NR} + l_4^{NR}$ and $V_P^{NM} = l_3^{NM}G_\infty^{NM} + l_4^{NM}$, respectively. Then, we have $V_P^{NR} - V_P^{NM} = \frac{1-16\varphi^2}{128}\left(\frac{1}{(\delta+1-\xi)} + \frac{1}{r}\right)\frac{\rho^2}{k_s}\left(\frac{(\theta+\lambda)^2}{\beta(r+\delta+1-\xi)}\right)^2$. Obviously, when $0 < \varphi < 1/4$, $V_P^{NR} > V_P^{NM}$; when $1/4 < \varphi < 1$, $V_P^{NR} < V_P^{NM}$.

In scenarios *NR* and *NM*, similarly, the manufacturer profits are $V_M^{NR} = l_1^{NR}G_\infty^{NR} + l_2^{NR}$ and $V_M^{NM} = l_1^{NM}G_\infty^{NM} + l_2^{NM}$, respectively. Then, we have $V_M^{NR} - V_M^{NM} = \frac{1-8(1-\varphi)\varphi}{128}\left(\frac{1}{(\delta+1-\xi)} + \frac{1}{r}\right)\left(\frac{\rho^2}{k_s}\right)\left(\frac{(\theta+\lambda)^2}{\beta(r+\delta+1-\xi)}\right)^2$. Obviously, when $0 < \varphi < \left(2-\sqrt{2}\right)/4$ or $\left(2+\sqrt{2}\right)/4 < \varphi < 1$, $V_M^{NR} > V_M^{NM}$; when $\left(2-\sqrt{2}\right)/4 < \varphi < \left(2+\sqrt{2}\right)/4$, $V_M^{NR} < V_M^{NM}$. □

**Proof of Theorem 3.** In scenario *BR*, we need to establish the existence of continuously differentiable value functions $V_M^{BR}$ and $V_P^{BR}$ to ensure that there is a unique solution $G(t)$ to Equation (3) and the *HJB* equations. The *HJB* equations for each player are as follows:

$$rV_M^{BR} = \max_w\left\{wD_f^{BR} + (1-\alpha)\Delta D_b^{BR} + \frac{\partial V_M^{BR}}{\partial G}(\rho s - \delta G)\right\} \tag{A23}$$

$$rV_P^{BR} = \max_{p,u,s}\left\{ (p-w)D_f^{BR} + \alpha\Delta D_b^{BR} - \frac{1}{2}k_u u^2 - \frac{1}{2}k_s s^2 + \frac{\partial V_P^{BR}}{\partial G}(\rho s - \delta G) - F \right\} \quad (A24)$$

Since $M$ is the leader, we first determine the first-order necessary conditions for scenario $BR$ as $p^{BR} = \frac{(\theta+\lambda)\sqrt{G}+\beta w}{2\beta}$, $s^{BR} = \frac{\rho}{k_s}\frac{\partial V_P^{BR}}{\partial G}$, and $u^{BR} = \frac{\alpha\Delta\varepsilon}{k_u}$. Substituting these into $M$'s $HJB$ Equation (A23), the following equation can be obtained:

$$rV_M^{BR} = \max_{w}\left\{ \frac{(\theta+\lambda)w\sqrt{G} - \beta w^2}{2} + (1-\alpha)\Delta\left(a + \frac{\alpha\Delta\varepsilon^2}{k_u}\right) + \frac{\partial V_M^{BR}}{\partial G}\left(\frac{\rho^2}{k_s}\frac{\partial V_P^{BR}}{\partial G} - \delta G\right) \right\} \quad (A25)$$

Thus, the necessary condition for $M$ is $w^{BR} = \frac{(\theta+\lambda)\sqrt{G}}{2\beta}$. And we obtain the optimal decisions of the players to the game as Equation (A26). □

$$\begin{cases} w^{BR} = \frac{(\theta+\lambda)\sqrt{G}}{2\beta} \\ p^{BR} = \frac{3(\theta+\lambda)\sqrt{G}}{4\beta} \\ s^{BR} = \frac{\rho}{k_s}\frac{\partial V_P^{BR}}{\partial G} \\ u^{BR} = \frac{\alpha\Delta\varepsilon}{k_u} \end{cases} \quad (A26)$$

Replacing the strategies of each player in Equations (A23) and (A24), we obtain the following:

$$rV_M^{BR} = \frac{(\theta+\lambda)^2}{8\beta}G + (1-\alpha)\Delta\left(a + \frac{\alpha\Delta\varepsilon^2}{k_u}\right) + \frac{\partial V_M^{BR}}{\partial G}\left(\frac{\rho^2}{k_s}\frac{\partial V_P^{BR}}{\partial G} - \delta G\right) \quad (A27)$$

$$rV_P^{BR} = \frac{(\theta+\lambda)^2}{16\beta}G + \alpha a\Delta + \frac{(\alpha\Delta\varepsilon)^2}{2k_u} + \frac{\rho^2}{2k_s}\left(\frac{\partial V_P^{BR}}{\partial G}\right)^2 - \delta G\frac{\partial V_P^{BR}}{\partial G} - F \quad (A28)$$

Based on the structure of (A27) and (A28), we conjecture the linear-valued functions, $V_M^{BR} = l_1^{BR}G + l_2^{BR}$ and $V_P^{BR} = l_3^{BR}G + l_4^{BR}$, where $l_1^{BR}$, $l_2^{BR}$, $l_3^{BR}$, and $l_4^{BR}$ are the constant parameters to be identified. Substituting $V_M^{BR}$ and $V_P^{BR}$ and their derivatives into Equations (A27) and (A28) we obtain the following:

$$r\left(l_1^{BR}G + l_2^{BR}\right) = \left(\frac{(\theta+\lambda)^2}{8\beta} - \delta l_1^{BR}\right)G + (1-\alpha)a\Delta + \frac{(1-\alpha)\alpha(\Delta\varepsilon)^2}{k_u} + \frac{\rho^2}{k_s}l_1^{BR}l_3^{BR} \quad (A29)$$

$$r\left(l_3^{BR}G + l_4^{BR}\right) = \left(\frac{(\theta+\lambda)^2}{16\beta} - \delta l_3^{BR}\right)G + \alpha a\Delta + \frac{(\alpha\Delta\varepsilon)^2}{2k_u} + \frac{\rho^2}{2k_s}\left(l_3^{BR}\right)^2 - F \quad (A30)$$

then we can identify the parameter values as follows:

$$\begin{cases} l_1^{BR} = \frac{(\theta+\lambda)^2}{8\beta(r+\delta)} \\ l_2^{BR} = \frac{1}{r}\left[(1-\alpha)a\Delta + \frac{(1-\alpha)\alpha(\Delta\varepsilon)^2}{k_u} + \frac{\rho^2}{2k_s}\left(\frac{(\theta+\lambda)^2}{8\beta(r+\delta)}\right)^2\right] \\ l_3^{BR} = \frac{(\theta+\lambda)^2}{16\beta(r+\delta)} \\ l_4^{BR} = \frac{1}{r}\left[a\alpha\Delta + \frac{(\alpha\Delta\varepsilon)^2}{2k_u} + \frac{\rho^2}{2k_s}\left(\frac{(\theta+\lambda)^2}{16\beta(r+\delta)}\right)^2 - F\right] \end{cases} \quad (A31)$$

which are all strictly positive; therefore, the results show the concave profit functions with respect to the players' decision variables and the existence of a unique equilibrium that maximizes the objective function. The optimal value functions then become

$$V_M^{BR} = \frac{(\theta+\lambda)^2 G}{8\beta(r+\delta)} + \frac{1}{r}\left[(1-\alpha)a\Delta + \frac{(1-\alpha)\alpha(\Delta\varepsilon)^2}{k_u} + \frac{\rho^2}{2k_s}\left(\frac{(\theta+\lambda)^2}{8\beta(r+\delta)}\right)^2\right]　\text{(A32)}$$

$$V_P^{BR} = \frac{(\theta+\lambda)^2 G}{16\beta(r+\delta)} + \frac{1}{r}\left[a\alpha\Delta + \frac{(\alpha\Delta\varepsilon)^2}{2k_u} + \frac{\rho^2}{2k_s}\left(\frac{(\theta+\lambda)^2}{16\beta(r+\delta)}\right)^2 - F\right]　\text{(A33)}$$

By solving the differential equation of Equation (3), we obtain the optimal brand goodwill trajectory as $G^{BR} = (G_0 - G_\infty^{BR})e^{-\delta t} + G_\infty^{BR}$, where $G_\infty^{BR} = \frac{\rho^2(\theta+\lambda)^2}{16\beta k_s(r+\delta)\delta}$, $G_\infty^{BR}$ is the steady-state goodwill.

Following the proof procedure of Theorems 1 and 2, the equilibrium strategies and E-CLSC members' profits under scenario $BM$ can be obtained, which will not be repeated here.

**Proof of Theorem 4.** In scenario $BM$, we need to establish the existence of continuously differentiable value functions $V_M^{BM}$ and $V_P^{BM}$ to ensure that there is a unique solution $G(t)$ to Equation (3) and the $HJB$ equations. The $HJB$ equations for each player are the following:

$$rV_M^{BM} = \max_p\left\{(1-\varphi)pD_f^{BM} + (1-\alpha)\Delta D_b^{BM} + \frac{\partial V_M^{BM}}{\partial G}(\rho s - \delta G)\right\}　\text{(A34)}$$

$$rV_P^{BM} = \max_{u,s}\left\{\varphi p(t)D_f^{BM} + \alpha\Delta D_b^{BM} - \frac{1}{2}k_u u^2 - \frac{1}{2}k_s s^2 + \frac{\partial V_P^{BM}}{\partial G}(\rho s - \delta G) - F\right\}　\text{(A35)}$$

Since $M$ is the leader, we first determine the first-order necessary conditions for scenario $BM$ as $s^{BM} = \frac{\rho}{k_s}\frac{\partial V_P^{BM}}{\partial G}$ and $u^{BM} = \frac{\alpha\Delta\varepsilon}{k_u}$. Substituting these into $M$'s $HJB$ Equation (A34), the following equation can be obtained:

$$rV_M^{BM} = \max_p\left\{(1-\varphi)p\left((\theta+\lambda)\sqrt{G} - \beta p\right) + (1-\alpha)\Delta\left(a + \frac{\alpha\Delta\varepsilon^2}{k_u}\right) + \frac{\partial V_M^{BM}}{\partial G}\left(\frac{\rho^2}{k_s}\frac{\partial V_P^{BM}}{\partial G} - \delta G\right)\right\}　\text{(A36)}$$

Thus, the necessary condition for $M$ is $p^{BM} = \frac{(\theta+\lambda)\sqrt{G}}{2\beta}$. And we obtain the optimal decisions of the players to the game as Equation (A37). □

$$\begin{cases} p^{BM} = \frac{(\theta+\lambda)\sqrt{G}}{2\beta} \\ s^{BM} = \frac{\rho}{k_s}\frac{\partial V_P^{BM}}{\partial G} \\ u^{BM} = \frac{\alpha\Delta\varepsilon}{k_u} \end{cases}　\text{(A37)}$$

Replacing the strategies of each player in Equations (A34) and (A35), we obtain the following:

$$rV_M^{BM} = \frac{(1-\varphi)(\theta+\lambda)^2}{4\beta}G + (1-\alpha)\Delta\left(a + \frac{\alpha\Delta\varepsilon^2}{k_u}\right) + \frac{\partial V_M^{BM}}{\partial G}\left(\frac{\rho^2}{k_s}\frac{\partial V_P^{BM}}{\partial G} - \delta G\right)　\text{(A38)}$$

$$rV_P^{BM} = \left(\frac{\varphi(\theta+\lambda)^2}{4\beta}\right)G + \alpha a\Delta + \frac{(\alpha\Delta\varepsilon)^2}{2k_u} + \frac{\rho^2}{2k_s}\left(\frac{\partial V_P^{BM}}{\partial G}\right)^2 - \delta G\frac{\partial V_P^{BM}}{\partial G} - F　\text{(A39)}$$

Based on the structure of (A38) and (A39), we conjecture the linear-valued functions, $V_M^{BM} = l_1^{BM}G + l_2^{BM}$ and $V_P^{BM} = l_3^{BM}G + l_4^{BM}$, where $l_1^{BM}$, $l_2^{BM}$, $l_3^{BM}$, and $l_4^{BM}$ are the

constant parameters to be identified. Substituting $V_M^{BM}$ and $V_P^{BM}$ and their derivatives into Equations (A38) and (A39), we obtain the following:

$$r\left(l_1^{BM}G + l_2^{BM}\right) = \left\{\left(\frac{(1-\varphi)(\theta+\lambda)^2}{4\beta} - \delta l_1^{BM}\right)G + (1-\alpha)a\Delta + \frac{(1-\alpha)\alpha(\Delta\varepsilon)^2}{k_u} + \frac{\rho^2}{k_s}l_1^{BM}l_3^{BM}\right. \tag{A40}$$

$$r\left(l_3^{BM}G + l_4^{BM}\right) = \left(\frac{\varphi(\theta+\lambda)^2}{4\beta} - \delta l_3^{BM}\right)G + \alpha a\Delta + \frac{(\alpha\Delta\varepsilon)^2}{2k_u} + \frac{\rho^2}{2k_s}\left(l_3^{BM}\right)^2 - F \tag{A41}$$

then we can identify the parameter values as follows:

$$\begin{cases} l_1^{BM} = \frac{(1-\varphi)(\theta+\lambda)^2}{4\beta(r+\delta)} \\ l_2^{BM} = \frac{1}{r}\left[(1-\alpha)\Delta a + \frac{(1-\alpha)\alpha(\Delta\varepsilon)^2}{k_u} + \frac{\rho^2(1-\varphi)\varphi}{k_s}\left(\frac{(\theta+\lambda)^2}{4\beta(r+\delta)}\right)^2\right] \\ l_3^{BM} = \frac{\varphi(\theta+\lambda)^2}{4\beta(r+\delta)} \\ l_4^{BM} = \frac{1}{r}\left[a\alpha\Delta + \frac{(\alpha\Delta\varepsilon)^2}{2k_u} + \frac{\rho^2}{2k_s}\left(\frac{\varphi(\theta+\lambda)^2}{4\beta(r+\delta)}\right)^2 - F\right] \end{cases} \tag{A42}$$

which are all strictly positive; therefore, the results show the concave profit functions with respect to the players' decision variables and the existence of a unique equilibrium that maximizes the objective function. The optimal value functions then become

$$V_M^{BM} = \frac{(1-\varphi)(\theta+\lambda)^2}{4\beta(r+\delta)}G + \frac{1}{r}\left[(1-\alpha)\Delta a + \frac{(1-\alpha)\alpha(\Delta\varepsilon)^2}{k_u} + \frac{\rho^2(1-\varphi)\varphi}{k_s}\left(\frac{(\theta+\lambda)^2}{4\beta(r+\delta)}\right)^2\right] \tag{A43}$$

$$V_P^{BM} = \frac{\varphi(\theta+\lambda)^2}{4\beta(r+\delta)}G + \frac{1}{r}\left[a\alpha\Delta + \frac{(\alpha\Delta\varepsilon)^2}{2k_u} + \frac{\rho^2}{2k_s}\left(\frac{\varphi(\theta+\lambda)^2}{4\beta(r+\delta)}\right)^2 - F\right] \tag{A44}$$

By solving the differential equation of Equation (3), we obtain the optimal brand goodwill trajectory as $G^{BM} = \left(G_0 - G_\infty^{BM}\right)e^{-\delta t} + G_\infty^{BM}$, where $G_\infty^{BM} = \frac{\varphi\rho^2(\theta+\lambda)^2}{4\beta k_s(r+\delta)\delta}$, $G_\infty^{BM}$ is the steady-state goodwill. Similarly, the equilibrium strategies and optimal value functions for each enterprise under scenario *BM* can be further obtained. The proof procedure is no different from the rest of Theorems 1, 2, and 3, and will not be repeated here.

**Proof of Corollary 2.** In scenarios *BR* and *BM*, the platform profits are $V_P^{BR} = l_3^{BR}G_\infty^{BR} + l_4^{BR}$ and $V_P^{BM} = l_3^{BM}G_\infty^{BM} + l_4^{BM}$, respectively. Then, we have $V_P^{BR} - V_P^{BM} = \frac{(1-16\varphi^2)}{16}\left(\frac{1}{\delta} + \frac{1}{2r}\right)\frac{\rho^2}{k_s}\left(\frac{(\theta+\lambda)^2}{4(r+\delta)\beta}\right)^2$. Obviously, when $0 < \varphi < 1/4$, $V_P^{BR} > V_P^{BM}$; when $1/4 < \varphi < 1$, $V_P^{BR} < V_P^{BM}$.

In scenarios *BR* and *BM*, similarly, the manufacturer profits are $V_M^{BR} = l_1^{BR}G_\infty^{BR} + l_2^{BR}$ and $V_M^{BM} = l_1^{BM}G_\infty^{BM} + l_2^{BM}$, respectively. Then, we have $V_M^{BR} - V_M^{BM} = \frac{1-8(1-\varphi)\varphi}{128}\left(\frac{1}{\delta} + \frac{1}{r}\right)\left(\frac{(\theta+\lambda)^2}{(r+\delta)\beta}\right)^2\left(\frac{\rho^2}{k_s}\right)$. Obviously, when $0 < \varphi < \left(2 - \sqrt{2}\right)/4$ or $\left(2 + \sqrt{2}\right)/4 < \varphi < 1$, $V_M^{BR} > V_M^{BM}$; when $\left(2 - \sqrt{2}\right)/4 < \varphi < \left(2 + \sqrt{2}\right)/4$, $V_M^{BR} < V_M^{BM}$. □

**Proof of Theorem 5.** In the reselling model, the comparative relationships between the quantity of used products recycling, steady-state goodwill, demand, equilibrium strategies, and manufacturer profits with and without blockchain technology are as follows:

(1) $D_b^{NR} - D_b^{BR} = (\xi - 1)a + \frac{(\xi^2-1)\alpha\Delta\varepsilon^2}{k_u}$, because $\xi < 1$, we have $D_b^{NR} < D_b^{BR}$

(2) $G_\infty^{NR} - G_\infty^{BR} = \left( \frac{1}{(\delta+1-\xi)(r+\delta+1-\xi)} - \frac{1}{\delta(r+\delta)} \right) \left( \frac{\rho^2}{k_s} \right) \frac{(\theta+\lambda)^2}{16\beta}$, because $\xi < 1$, we have $G_\infty^{NR} < G_\infty^{BR}$.

(3) $D_f^{NR} - D_f^{BR} = \frac{(\theta+\lambda)\left( \sqrt{G_\infty^{NR}} - \sqrt{G_\infty^{BR}} \right)}{4}$, because $G_\infty^{NR} < G_\infty^{BR}$, we have $D_f^{NR} < D_f^{BR}$.

(4) $w_\infty^{NR} - w_\infty^{BR} = \frac{(\theta+\lambda)\left( \sqrt{G_\infty^{NR}} - \sqrt{G_\infty^{BR}} \right)}{2\beta}$, because $G_\infty^{NR} < G_\infty^{BR}$, we have $w_\infty^{NR} < w_\infty^{BR}$.

(5) $p_\infty^{NR} - p_\infty^{BR} = \frac{3(\theta+\lambda)\left( \sqrt{G_\infty^{NR}} - \sqrt{G_\infty^{BR}} \right)}{4\beta}$, because $G_\infty^{NR} < G_\infty^{BR}$, we have $p_\infty^{NR} < p_\infty^{BR}$.

(6) $s^{NR} - s^{BR} = \left( \frac{1}{(r+\delta+1-\xi)} - \frac{1}{(r+\delta)} \right) \frac{\rho(\theta+\lambda)^2}{16k_s\beta}$, because $\xi < 1$, we have $s^{NR} < s^{BR}$.

(7) $u^{NR} - u^{BR} = \frac{(\xi-1)\alpha\Delta\varepsilon}{k_u}$, because $\xi < 1$, we have $u^{NR} < u^{BR}$.

(8) $V_M^{NR} - V_M^{BR} = \left\{ \begin{array}{l} \left[ \frac{1}{(\delta+1-\xi)} \left( \frac{1}{\beta(r+\delta+1-\xi)} \right)^2 - \frac{1}{\delta} \left( \frac{1}{(r+\delta)\beta} \right)^2 \right] \frac{\rho^2(\theta+\lambda)^4}{128k_s} \\ + \frac{1}{r} \left[ (\xi-1)(1-\alpha)a\Delta + \frac{(\xi^2-1)(1-\alpha)\alpha(\Delta\varepsilon)^2}{k_u} + \left( \frac{1}{(r+\delta+1-\xi)^2} - \frac{1}{(r+\delta)^2} \right) \frac{\rho^2}{2k_s} \left( \frac{(\theta+\lambda)^2}{8\beta} \right)^2 \right] \end{array} \right\}$,

because $\xi < 1$, we have $V_M^{NR} < V_M^{BR}$.

We omit the proof in the marketplace model since the proof procedure is similar. □

**Proof of Theorem 6.** In the reselling model, the blockchain deployment conditions need to be satisfied: $V_P^{BR} > V_P^{NR}$, where $V_P^{BR} = l_3^{BR}G_\infty^{BR} + l_4^{BR}$ and $V_P^{NR} = l_3^{NR}G_\infty^{NR} + l_4^{NR}$, respectively. Thus, we have $V_P^{BR} - V_P^{NR} = \frac{1}{2k_s} \left( \frac{\rho(\theta+\lambda)^2}{16\beta} \right)^2 \left[ \frac{2r+\delta}{(r+\delta)^2\delta} - \frac{2r+\delta+1-\xi}{(r+\delta+1-\xi)^2(\delta+1-\xi)} \right] + (1-\xi)a\alpha\Delta + \frac{(1-\xi^2)(\alpha\Delta\varepsilon)^2}{2k_u} - F > 0$, i.e., $F < \frac{1}{2k_s} \left( \frac{\rho(\theta+\lambda)^2}{16\beta} \right)^2 \left[ \frac{2r+\delta}{(r+\delta)^2\delta} - \frac{2r+\delta+1-\xi}{(r+\delta+1-\xi)^2(\delta+1-\xi)} \right] + (1-\xi)a\alpha\Delta + \frac{(1-\xi^2)(\alpha\Delta\varepsilon)^2}{2k_u}$. □

In the marketplace model, the blockchain deployment conditions need to be satisfied: $V_P^{BM} > V_P^{NM}$, where $V_P^{BM} = l_3^{BM}G_\infty^{BM} + l_4^{BM}$ and $V_P^{NM} = l_3^{NM}G_\infty^{NM} + l_4^{NM}$, respectively. Thus, we have $V_P^{BM} - V_P^{NM} = \frac{1}{2k_s} \left( \frac{\varphi\rho(\theta+\lambda)^2}{4\beta} \right)^2 \left( \frac{2r+\delta}{(r+\delta)^2\delta} - \frac{2r+\delta+1-\xi}{(r+\delta+1-\xi)^2(\delta+1-\xi)} \right) + (1-\xi)a\alpha\Delta + \frac{(1-\xi^2)(\alpha\Delta\varepsilon)^2}{2k_u} - F > 0$, i.e., $F < \frac{1}{2k_s} \left( \frac{\varphi\rho(\theta+\lambda)^2}{4\beta} \right)^2 \left( \frac{2r+\delta}{(r+\delta)^2\delta} - \frac{2r+\delta+1-\xi}{(r+\delta+1-\xi)^2(\delta+1-\xi)} \right) + (1-\xi)a\alpha\Delta + \frac{(1-\xi^2)(\alpha\Delta\varepsilon)^2}{2k_u}$.

**Proof of Theorem 7.** According to the proof process of Corollaries 1 and 2, we obtain the sales model selection for the platform and manufacturer's intention to collaborate in Theorem 7. □

**Proof of Theorem 8.** When the online platform collaborates with a blockchain provider, there is a per-unit usage cost $b$. The proof process is consistent with the process of Theorems 1, 2, 3, and 4 and will not be expanded here. We can obtain the following:

In scenario $BR$, the optimal brand trajectory is $G^{BR} = \left( G_0 - G_\infty^{BR} \right)e^{-\delta t} + G_\infty^{BR}$, and its steady-state $G_\infty^{BR} = \frac{\rho^2(\theta+\lambda)^2}{16\beta k_s(r+\delta)\delta}$. The E-CLSC members' equilibrium strategies are $w_\infty^{BR} = \frac{(\theta+\lambda)\sqrt{G_\infty^{BR}}}{2\beta}$, $p_\infty^{BR} = \frac{3(\theta+\lambda)\sqrt{G_\infty^{BR}}}{4\beta}$, $s^{BR} = \frac{\rho(\theta+\lambda)^2}{16\beta k_s(r+\delta)}$, $u^{BR} = \frac{\alpha(\Delta-b)\varepsilon}{k_u}$. □

The optimal profits for the manufacturer and the platform are $V_M^{BR} = \frac{1}{2k_s\delta 8} \left( \frac{\rho(\theta+\lambda)^2}{8\beta(r+\delta)} \right)^2 + \frac{1}{r} \left[ (1-\alpha)a(\Delta-b) + \frac{(1-\alpha)\alpha((\Delta-b)\varepsilon)^2}{k_u} + \frac{\rho^2}{2k_s} \left( \frac{(\theta+\lambda)^2}{8\beta(r+\delta)} \right)^2 \right]$ and $V_P^{BR} = \frac{1}{\delta k_s} \left( \frac{\rho(\theta+\lambda)^2}{16\beta(r+\delta)} \right)^2 + \frac{1}{r} \left[ a\alpha(\Delta-b) + \frac{(\alpha(\Delta-b)\varepsilon)^2}{2k_u} + \frac{\rho^2}{2k_s} \left( \frac{(\theta+\lambda)^2}{16\beta(r+\delta)} \right)^2 \right]$, respectively.

In scenario $BM$, the optimal brand trajectory is $G^{BM} = \left(G_0 - G_\infty^{BM}\right)e^{-\delta t} + G_\infty^{BM}$, and its steady-state $G_\infty^{BM} = \frac{\varphi\rho^2(\theta+\lambda)^2}{4\beta k_s(r+\delta)\delta}$. The E-CLSC members' equilibrium strategies are $p_\infty^{BM} = \frac{(\theta+\lambda)\sqrt{G_\infty^{BM}}}{2\beta}$, $s^{BM} = \frac{\varphi\rho(\theta+\lambda)^2}{4\beta k_s(r+\delta)}$, $u^{BM} = \frac{\alpha(\Delta-b)\varepsilon}{k_u}$.

The optimal profits for the manufacturer and the platform are
$$V_M^{BM} = \frac{\varphi(1-\varphi)}{\delta k_s}\left(\frac{\rho(\theta+\lambda)^2}{4\beta(r+\delta)}\right)^2 + \frac{1}{r}\left[(1-\alpha)(\Delta-b)a + \frac{(1-\alpha)\alpha((\Delta-b)\varepsilon)^2}{k_u} + \frac{\rho^2(1-\varphi)\varphi}{k_s}\left(\frac{(\theta+\lambda)^2}{4\beta(r+\delta)}\right)^2\right]$$
and $V_P^{NM} = \frac{1}{\delta k_s}\left(\frac{\varphi\rho(\theta+\lambda)^2}{4\beta(r+\delta)}\right)^2 + \frac{1}{r}\left[a\alpha(\Delta-b) + \frac{(\alpha(\Delta-b)\varepsilon)^2}{2k_u} + \frac{\rho^2}{2k_s}\left(\frac{\varphi(\theta+\lambda)^2}{4\beta(r+\delta)}\right)^2\right]$, respectively.

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
