# Peer review of "The Impact of Blockchain Technology Adoption on an E-Commerce Closed-Loop Supply Chain Considering Consumer Trust"

_sustainability, doi:10.3390/su16041535_

Round 1
Reviewer 1 Report
Comments and Suggestions for Authors
Th

must be improved
Reviewer 2 Report
Comments and Suggestions for Authors
In this paper, the authors investigate the value of blockchain technology in building consumer trust in recyclers. It is shown that using blockchain there is reliable proof of the sustainable behavior of recycling chain members. Formulas need more clarification. For instance, in line 419 eu is used, but above you used u(t). What is the difference between the two? Introduction is too long too. Parts of it could be moved to section 2.
Reviewer 3 Report
Comments and Suggestions for Authors
The paper deals with an important topic to which the attention of society should be directed. The main question addressed by this paper is the value of blockchain technology used in e-commerce convincing consumers to recycle. Papers focused on environmental issues are extremely important nowadays and therefore this paper is relevant for Sustainability journal and it could be interesting for its readers. The paper explores new areas or combines previous research topics such as strategic management, supply chain management, and online platforms together as it is explained in the part Literature review. The paper explains possible strengths and weaknesses of each scenario how the product can reach the consumer originally and the ways how to use online platform supporting the recycling activities of consumer. The role of consumers in recycling activities is essential. It proves that the blockchain technology can influence this process significantly and affect the decision making and effort to recycle. The paper is well structured and therefore it is readable but unfortunately the text is extremely long which can discourage readers. The readers can be unfortunately discouraged because of the papers length and the amount of formulas applied and many symbols included which make orientation and comprehensibility more difficult. The paper does not contain any empirical results and therefore recommendations and conclusions are based on the theoretical approach. Conclusions just highlight the main findings of the paper, possible managerial applications, and future ways of research. The conclusion is in favor of the blockchain technology on which the paper focused from the beginning to the end. Conclusions should contain the paper limitations which have not been addressed in the current version.
The following text contains recommendations how to improve the paper before its publishing. Authors should also focus on the issues pointed out in the previous paragraph as simplicity, reader's attractiveness, and missing paper's limitations.
There should be a space after the comma, even when references are mentioned in the text e.g. lines 67, 87, 12, 188, 192, 204, 416, 935.
The paper template should be checked if this specific journal does not require to cite in the text not only the author/s but also the year of paper publishing.
The text should not contain when the webpages were accessed. This piece of information should be included in the section of references not in the text such as e.g. lines 97, 102, 110, 466, 501.
Tables should not break on two pages as Tables 1 and 2.
It would look more professional if the identification of paper in Table 1 would not break on two lines.
Table of Figure title should not be placed on the page without the object itself – e.g. lines 543, 953.
There should not be space between sentence commas and dots such as lines 631, 634.
Chapters should not start with the figure but by the introductory paragraph as the chapters 6.2, 6.4, 6.5, 6.6.
There should be spaces around =, not the of the line 859.
Figures 8 and 9 contain lines which are not parallel.
Comments on the Quality of English LanguageThere are only minor recommendations in the area of language. Their list follows.
Title of the paper should be written in large initial letters of the words.
Personal nouns as we and our should be avoided or their occurrence in the text should be limited. Scientific papers mostly avoid their presence at least in the abstract.
Scientific papers should be gender neutral. It is surprising that manufacturer is indicated as he and online platform as she – first occurrence – line 394. Line 520 mentions other possibility how to refer to these entities as P and M which are not repeatedly applied in the text. Repeated use of he/she indication in the paragraphs represented by lines 716-738, 793-804, 806-834, line 912.
Scientific papers should avoid the usage of the shortening – e.g. line 756 haven t.
Reviewer 4 Report
Comments and Suggestions for Authors
The article is devoted to improving online business processes in the field of waste processing based on the use of block chain technology.
As the authors of the article rightly note, the idea of ​ ​ ensuring user trust based on block chain technology is not new, but very promising. Block chain technology allows you to ensure reliable control of the integrity of transactions at all stages of business waste processing processes. This not only increases user confidence, but also opens up new promising opportunities for flexible business process management based on differential game theory.
The authors conducted a fairly complete analysis of known information sources. The shortcomings of some well-known scientific papers were accurately indicated.
Mathematical calculations are made correctly with the provision of detailed evidence. The drawings are accurate and informative. The results obtained are well illustrated.
The reference list contains the necessary relevant reference information sources.
The article is well subtracted and prepared. Of course, such an article should be recommended for publication in the journal Sustainability.
I think that this article may well take part in the competition for one of the best publications of 2024.
I congratulate the authors on a very good result and wish further scientific achievements.
